# In Silico Assessment of Class I Antiarrhythmic Drug Effects on *Pitx2*-Induced Atrial Fibrillation: Insights from Populations of Electrophysiological Models of Human Atrial Cells and Tissues

**DOI:** 10.3390/ijms22031265

**Published:** 2021-01-27

**Authors:** Jieyun Bai, Yijie Zhu, Andy Lo, Meng Gao, Yaosheng Lu, Jichao Zhao, Henggui Zhang

**Affiliations:** 1Department of Electronic Engineering, College of Information Science and Technology, Jinan University, Guangzhou 510632, China; zyj1934261010@stu2019.jnu.edu.cn; 2Auckland Bioengineering Institute, University of Auckland, Auckland 1010, New Zealand; alo006@auckland.ac.nz (A.L.); j.zhao@auckland.ac.nz (J.Z.); 3Department of Computer Science and Technology, College of Electrical Engineering and Information, Northeast Agricultural University, Harbin 150030, China; 4Biological Physics Group, School of Physics and Astronomy, The University of Manchester, Manchester M13 9PL, UK; henggui.zhang@manchester.ac.uk

**Keywords:** action potential, atrial fibrillation, in silico model, population of models, class I antiarrhythmic drugs, flecainide, disopyramide, quinidine, propafenone, Pitx2

## Abstract

Electrical remodelling as a result of homeodomain transcription factor 2 (Pitx2)-dependent gene regulation was linked to atrial fibrillation (AF) and AF patients with single nucleotide polymorphisms at chromosome 4q25 responded favorably to class I antiarrhythmic drugs (AADs). The possible reasons behind this remain elusive. The purpose of this study was to assess the efficacy of the AADs disopyramide, quinidine, and propafenone on human atrial arrhythmias mediated by Pitx2-induced remodelling, from a single cell to the tissue level, using drug binding models with multi-channel pharmacology. Experimentally calibrated populations of human atrial action po-tential (AP) models in both sinus rhythm (SR) and Pitx2-induced AF conditions were constructed by using two distinct models to represent morphological subtypes of AP. Multi-channel pharmaco-logical effects of disopyramide, quinidine, and propafenone on ionic currents were considered. Simulated results showed that Pitx2-induced remodelling increased maximum upstroke velocity (dVdtmax), and decreased AP duration (APD), conduction velocity (CV), and wavelength (WL). At the concentrations tested in this study, these AADs decreased dVdtmax and CV and prolonged APD in the setting of Pitx2-induced AF. Our findings of alterations in WL indicated that disopyramide may be more effective against Pitx2-induced AF than propafenone and quinidine by prolonging WL.

## 1. Introduction

Although atrial fibrillation (AF) incidence increases with age, and with the context of concomitant cardiac pathologies [1], population-based genome-wide association studies (GWASs) showed that one-third of AF patients carry common genetic variants, suggesting that AF has a heritable component [2]. Recently, many AF-associated loci were identified in GWASs [3,4,5,6,7,8,9] and the most common AF susceptibility locus first identified in European, Chinese, and Japanese populations is located on chromosome 4q25 [10]. The gene-poor 4q25 region harbors paired-like homeodomain transcription factor 2 (*Pitx2*), which has been fundamentally linked to AF [11,12,13,14,15,16,17,18,19,20], although the basis for this connection remains obscure. In early cardiac embryogenesis, *Pitx2* suppresses left atrial automaticity and the formation of “sinus node-like structures” in the left atrium [21] and contributes to the formation of the pulmonary vein myocardium [13]. In the adult heart, *Pitx2* is mainly expressed in the left atrium and pulmonary vein [11]. Experimental studies of *Pitx2*-induced AF have indicated that the downregulation of Pitx2 creates a predisposition to AF without marked structural changes in the atria [11,15,18,22] via shortened atrial repolarisation [22], a more depolarised resting membrane potential (RMP) [18], and abnormalities in calcium cycling [17,23,24]. Gene expression analyses highlighted that *Pitx2* regulated genes of ion channels and gap junctions [11,12,17,22,23,25,26] in a dose-dependent manner. Based on these experimental data on changes in the expression of ion channels and gap junctions, we constructed multi-scale models of human atrial electrophysiology to investigate mechanisms by which *Pitx2*-induced remodelling promotes AF in our previous studies [27,28,29,30,31,32,33]. However, the effective management of AF remains a challenge, and is incompletely understood in the context of Pitx2-induced AF.

A population-based study assessed the influence of AF-associated loci on the response to antiarrhythmic drug (AAD) therapies and showed that carriers of the variant allele at rs10033646 on chromosome 4q25 responded favorably to class I AADs [34]. Class I AADs used in AF include flecainide, disopyramide, quinidine, and propafenone [35]. The pharmacological effects of flecainide in Pitx2-induced AF were investigated by using a multi-scale computational model and simulated results demonstrated that flecainide is effective for the treatment of Pitx2-induced AF patients by preventing spontaneous calcium release [36]. However, the efficacy of other class I AADs, disopyramide, quinidine, and propafenone, on human atrial arrhythmias mediated by *Pitx2*-induced remodelling remains elusive.

Although the efficacy of disopyramide, quinidine, and propafenone on human atrial patho-electrophysiology associated with human Ether-à-go-go-Related Gene(hERG) -linked short QT syndrome has been investigated using a multi-scale computational model [37], mechanisms by which short QT mutations promote AF may be different from that underlying *Pitx2*-induced AF [27,28,29,30,31,32,36]. Additionally, both clinical [34] and theoretical [36] studies of pharmacotherapy for *Pitx2*-induced AF often ignored inter-subject variability in atrial electrophysiology properties. Population-based computational approaches that can capture key disease conditions have proven valuable for understanding inter-subject variability in electrophysiological properties [38,39,40] and cardiotoxicity [41,42,43,44,45,46,47,48,49,50,51,52]. Similarly, these methods were useful in the study of AF [53,54,55,56,57]. Recently, a novel Quantitative Systems Pharmacology Framework was proposed based on these population-based computational models and the effects of a multi-atrial-predominant potassium current block in AF were investigated [54].

Here, following the Quantitative Systems Pharmacology Framework developed by Ni et al. [54], we constructed populations of in silico models calibrated to values of action potential (AP) biomarkers reported in an experimental dataset on AP recordings [58]. Using these experimentally calibrated models, we simulated and assessed actions of class I AADs on human atrial electrophysiology. AP duration (APD) and maximum upstroke velocity (dVdt_max_) were quantified to evaluate anti-AF effects of class I AADs on AP at the cellular level, while conduction velocity (CV) and wavelength (WL) were quantified to assess the effects of class I AADs on AP propagation at the tissue level. Sensitivity analyses of AP biomarkers were applied to understand the ionic mechanisms underlying *Pitx2*-induced AF and the efficacy of these AADs. Finally, we performed population-based simulations of *Pitx2*-induced remodelling and predicted a reduction in APD, CV, and WL and an increase in dVdt_max_. Further simulations of actions of class I AADs on *Pitx2*-induced AF exhibited APD prolongation and a reduction in CV and dVdt_max_. Our results showed that disopyramide led to WL prolongation compared to the drug-free AF conditions. These findings suggest that disopyramide may be effective against *Pitx2*-induced AF.

## 2. Results

### 2.1. Overview of In Silico Assessment of Class I Antiarrhythmic Drugs

A flow chart illustrating the process for the in silico assessment of class I AADs effects on *Pitx2*-induced AF is presented in Figure 1. First, the initial population of sinus rhythm (SR) models was created by randomly perturbing parameters associated with ionic properties (Table 1) of the baseline human atrial cell models [41]. The Bai et al. model displaying a type-1 AP with notch-and-dome morphology [27] and the Grandi et al. model displaying a type-3 AP with a typical triangular shape [59] were chosen as the baseline human atrial cell models to represent distinct morphological subtypes of human atrial AP [60]. Parameters associated with ionic properties were allowed to vary independently according to a log-normal distribution and sigma was set to be 0.2 to cover a range of variability similar to that seen in experiments based on previous studies [41,54,61,62]. Second, we used the initial models to simulate human atrial AP by considering stimulation frequency (1 Hz) under the experimental conditions [58] and calculated the AP biomarkers of each initial model. The initial models generated in the previous step were selected to constitute the SR population whose simulated electrophysiological properties are in range with the same properties in experimental data on AP biomarkers (including dVdt_max_, RMP, APD_50_, and APD_90_) in Table 2. This step yields the experimentally calibrated population of SR models [53]. Third, electrical remodelling due to impaired *Pitx2* (Table 3) was introduced into SR model variants to generate the initial population of AF models. Fourth, we used experimental AP biomarkers to calibrate the AF population by excluding the model variants in which values of simulated biomarkers were outside the experimentally observed range (Table 3) reported by Ravens et al [58]. Fifth, the blocking effects of AADs on ion channels (Table 4) were incorporated into experimentally calibrated Pitx2-induced AF models to evaluate their effects on the virtual atrial myocytes. Finally, sensitivity analyses [41,54] of dVdt_max_ and APD_90_ at the cellular level and CV and WL at the tissue level were applied to understand modulations by ionic parameters of *Pitx2*-induced remodelling and ion channels affected by AADs. On the basis of model responses, AADs that eliminated the arrhythmogenic propensity of the atrial substrate arising from *Pitx2*-induced remodelling were selected.

### 2.2. The Experimentally Calibrated Populations of Human Atrial Myocytes under SR and AF Conditions

Representing a type-1 AP with notch-and-dome morphology, the Bai et al. model [27] was used to generate the initial SR population of human atrial cell model variants. According to the physiological range of AP biomarkers (dVdt_max_, RMP, APD_50_, and APD_90_ values at 1 Hz pacing) measured experimentally (Table 2), populations of 745 SR models out of the initial pool of 1200 models were selected to generate the experimentally calibrated population of SR models. Next, we incorporated *Pitx2*-induced electrical remodelling (Table 3) into the SR population of 745 variants to generate the initial AF population, and then calibrated this AF population to the experimentally measured AP biomarkers (Table 2). Representative AP traces for the experimentally calibrated SR population of 745 models and AF population of 621 models, respectively, are shown in Figure 2a,b. Distributions of AP biomarkers of the experimentally calibrated SR models are compared to that of AF models (Figure 2c,d). In detail (Table 5), dVdt_max_ is increased from 197.87 ± 83.18 V/s for the SR condition to 219.21 ± 90.48 V/s for the AF condition, whereas APD_90_ (250.10 ± 41.59 ms vs. 181.93 ± 23.66 ms) and APD_50_ (187.84 ± 32.36 ms vs. 121.22 ± 24.13 ms) were abbreviated due to *Pitx2*-induced remodelling. No significant changes in RMP (−79.28 ± 3.29 mV vs. −79.81 ± 3.22 mV) were observed.

Representing a type-3 AP with a typical triangular shape, the Grandi et al. model [59] was also used to generate the experimentally calibrated populations according to the physiological range of AP biomarkers [58] measured experimentally (Table 2). One hundred and seventy SR models and 87 AF models were selected to constitute the SR population and the AF population, respectively. Figure 3 shows representative AP traces (Figure 3a,b) and distributions of AP biomarkers (Figure 3c,d) for the SR population and the AF population. Consistent with changes in AP biomarkers observed in the Bai et al. model (Figure 2c,d and Table 5), in particular, dVdt_max_ is increased from 351.14 ± 48.90 V/s for the SR population to 368.66 ± 52.15 V/s for the AF population, whereas APD_90_ is decreased from 251.08 ± 61.50 ms for the SR population to 241.82 ± 37.98 ms for the AF population (Table 5).

### 2.3. Sensitivity Analysis Revealed Alterations in Depolarisation and Repolarisation of AP

Experimental studies have demonstrated that sensitivity analyses of dVdt_max_ and APD_90_ can provide insights into alterations in the depolarisation rate [80,81,82,83] and the repolarisation time [84] of the cardiac tissue. We applied partial correlation analysis to predict alterations in dVdt_max_ and APD_90_ when electrical remodelling (listed in Table 3) due to impaired Pitx2 was introduced or ion channels (listed in Table 4) were blocked by class I AADs. Parameters associated with Pitx2-induced electrical remodelling include G_Na_, G_CaL_, G_Ks_, G_K1_, G_rel_, and G_up_, while parameters affected by actions of AADs (including disopyramide, quinidine, and propafenone) include G_Na_, G_CaL_, G_to_, G_Ks_, G_Kr_, G_Kur_, G_KATP_, G_KAch,Ado_, and G_K1_. We used partial correlation analysis to calculate partial correlation coefficients (PCCs) to quantify correlations between the parameter values and dVdt_max_ and APD_90_. In Figure 4a,d, dVdt_max_ PPCs are positive for G_Na_ in Bai et al. and Grandi et al. populations, suggesting that upregulated I_Na_ in *Pitx2*-induced remodelling would tend to increase the depolarisation rate and inhibit I_Na_ due to actions of AADs reducing depolarisation rate. In Figure 4e,h, APD_90_ PPCs are negative for G_Ks_, G_Kr_, G_KAch,Ado_, and G_K1_ but not for G_CaL_, indicating that upregulated G_Ks_ and downregulated G_CaL_ in *Pitx2*-induced remodelling cause APD shortening and blocking these potassium currents (i.e., I_Ks_, I_Kr_, I_KAch_, and I_K1_) would tend to prolong APD. APD_90_ PPCs for G_to_, G_KATP_, and G_Kur_ are also negative in Grandi et al. but not Bai et al. populations. Therefore, in Grandi et al. populations, APD_90_ PPCs for G_to_, G_Ks_, G_Kr_, G_Kur_, G_KAch,Ado_, G_KATP_, and G_K1_ are negative, suggesting that inhibiting these currents due to actions of AADs produces positive APD prolongation. In contrast, in Bai et al. populations, negative APD_90_ PPCs are seen for I_Ks_, I_Kr_, I_KAch_, and I_K1_ only, indicating that the blocking effects of class I drugs on I_Ks_, I_Kr_, I_KAch_, and I_K1_ are associated with positive APD prolongation, but the blocking effects of AADs on G_to_, G_KATP_, and G_Kur_ with negative APD prolongation.

### 2.4. Antiarrhythmic Effects of Class I Drugs on dVdt_max_ and APD_90_ at the Cellular Level

According to experimental data on actions of class I AADs on ion currents, their effects were incorporated into experimentally calibrated models in the AF population created with the Bai et al. model. Parameters associated with the effects of AADs were G_Na_, G_CaL_, G_to_, G_Ks_, G_Kr_, G_KATP_, G_KAch,Ado_, G_Kur_, and G_K1_ (Table 4). The class I AADs investigated here included disopyramide, quinidine, and propafenone. Taking into account plasma protein binding, estimates of the most likely unbound concentrations of propafenone, disopyramide, and quinidine have been given as ~0.15–1 μM [85,86,87], 1 μM, and 2 μM [88], respectively. To encompass the likely total as well as unbound concentrations, we chose to simulate the effects of a wide range of concentrations of propafenone (low dose Prop_L: 0.2, medium dose Prop_M: 0.5, and high dose Prop_H: 0.8 μM), disopyramide (Diso_L: 1.0, Diso_M: 2.0, and Diso_H: 5.0 μM), and quinidine (Quin_L: 1.0, Quin_M: 2.0, and Quin_H: 5.0 μM) [37,89]. Figure 5 shows the actions of disopyramide, quinidine, and propafenone on dVdt_max_ and APD_90_ in the AF condition. All drugs also prolonged APD_90_ in a dose-dependent manner (Figure 5a), with quinidine producing a slightly larger increase in APD_90_ across all concentrations investigated. In detail (Table 6), quinidine prolonged APD_90_ from 181.93 ± 23.66 ms in the drug-free AF condition to 199.32 ± 28.53 ms (Quin_L), 206.57 ± 32.27ms (Quin_M), and 219.08± 36.78 ms (Quin_H), whereas values of APD_90_ are 188.29 ± 24.57 ms (Diso_L), 192.17 ± 26.09 ms (Diso_M), and 199.32 ± 28.53 ms (Diso_H) upon application of disopyramide, and are 196.69 ± 27.72 ms (Prop_L), 205.60 ± 32.13 ms (Prop_M), and 210.37± 34.46 ms (Prop_H) upon application of propafenone. APD_90_ upon application of 5μM quinidine (Quin_H) in the AF condition is close to that in the drug-free SR condition (219.08 ± 36.78 vs. 250.10 ± 41.59). It can be seen in Figure 5b that disopyramide, quinidine, and propafenone reduced dVdt_max_ in a dose-dependent manner, with disopyramide and quinidine decreasing dVdt_max_ to a greater extent than propafenone. Values of dVdt_max_ upon application of quinidine and propafenone were smaller than those in the drug-free AF condition, but were larger than those in the drug-free SR condition. However, values of dVdt_max_ upon the application of disopyramide were smaller than those under drug-free AF and SR conditions.

Based on the experimentally calibrated AF models created with the Grandi et al. model, the antiarrhythmic effects of disopyramide, quinidine, and propafenone on dVdt_max_ and APD_90_ were investigated. Figure 5 shows all drugs prolonged APD_90_ (Figure 5c) and decreased dVdt_max_ (Figure 5d) in a dose-dependent manner. It can also be seen in Figure 5c that APD_90_ upon the application of disopyramide in the AF condition was larger than that in the drug-free AF condition, whereas values of APD_90_ upon the application of propafenone and quinidine in the AF condition were smaller than that in the drug-free AF condition. Quinidine decreased dVdt_max_ to a greater extent than disopyramide and propafenone. The value of dVdt_max_ upon the application of quinidine (Quin_H) was smaller than those under drug-free AF and SR conditions.

Together, our models developed by the Bai et al. model or the Grandi et al. model predicted that all drugs reduced dVdt_max_ and APD_90_ to different extents.

### 2.5. Antiarrhythmic Effects of Class I Drugs on CV and WL at the Tissue Level

To evaluate the effects of class I drugs on CV and WL, we constructed one-dimensional (1D) models of human atrial strands with the Bai et al. model to investigate the responses of electrical waves to AADs in tissue. Figure 6 shows CV and WL upon the application of disopyramide, quinidine, and propafenone in the AF condition, compared with those in the drug-free SR and AF conditions. CV was decreased from 0.56 ± 0.08 mm/ms for the SR condition to 0.45 ± 0.11 mm/ms for the AF condition. It can be seen in Figure 6a that disopyramide, quinidine, and propafenone reduced CV in a dose-dependent manner, with disopyramide decreasing CV to a greater extent than quinidine and propafenone. Figure 6b shows values of WL under drug-free SR and AF conditions, and upon the application of various concentrations of disopyramide, quinidine, and propafenone. WL was decreased from 139.62 ± 25.56 mm for the drug-free SR condition to 81.13 ± 18.91 mm for the drug-free AF condition. Compared with the drug-free AF condition, all drugs prolonged WL in a dose-dependent manner, with quinidine and propafenone increasing WL to a greater extent than disopyramide. In order of effects of AADs on WL, these drugs are quinidine, propafenone, and disopyramide.

Using 1D models of human atrial fibers developed with the Grandi et al. model, the antiarrhythmic effects of class I drugs on CV and WL were also investigated. All drugs decreased CV in a dose-dependent manner, with quinidine (Quin_H) decreasing CV to a greater extent than disopyramide and propafenone (Figure 6c). Figure 6d shows that disopyramide prolonged WL in a dose-dependent manner, whereas quinidine and propafenone shortened WL in a dose-dependent manner. In order of effects of AADs on WL, these drugs are disopyramide, propafenone, and quinidine.

Together, our models developed by the Bai et al. model or the Grandi et al. model predicted that all drugs reduced CV to different extents. However, only disopyramide can prolong WL in the *Pitx2*-induced AF condition. Quantitative summaries of the effects of class AADs on human atrial electrical activity in the *Pitx2*-induced AF condition are listed in Table 6 for the Bai et al. model and Table 7 for the Grandi et al. model. Alterations of electrophysiological properties are listed in Table 8.

## 3. Discussion

Population-based studies have assessed the influence of common single-nucleotide polymorphisms related to AF on the response to AAD therapies and showed that carriers of the variant allele at rs10033646 on chromosome 4q25 responded favorably to the class I AADs [34]. The actions of the class I AADs disopyramide, quinidine, and propafenone were assessed in the context of *Pitx2*-induced AF using a population-based Quantitative Systems Pharmacology Framework [54]. Through sensitivity and statistical analyses of our atrial cell and tissue populations, we found that WL prolongation could be achieved by disopyramide. This study provides clinically relevant insights into the pharmacology of WL prolongation by evaluating and comparing the actions of all three drugs in the context of *Pitx2*-induced AF, offering an important step toward in silico optimization of pharmacological therapy in this context.

### 3.1. Main Findings

The major findings presented in this study are as follows. (1) Populations of models based on two human atrial AP models are able to mimic a wide range of inter-subject variability in human atrial AP properties, as exhibited in a set of AP measurements from over 379 SR and AF patients. (2) *Pitx2*-induced remodelling (as reported in [36]) predicts abbreviated APD_90_ and increased dVdt_max_ at the cellular level, and decreased CV and shortened WL at the tissue level in SR versus AF conditions, as reported in experimental studies [17,58] (Table 2). (3) AP biomarkers (namely, APD_90_ and dVdt_max_) are correlated in both SR and AF cardiomyocytes created with the Bai et al. and the Grandi et al. models. dVdt_max_ is primarily determined by G_Na_, whereas APD_90_ is determined by G_CaL_, G_Ks_, G_Kr_, G_KAch,Ado_, and G_K1_. (4) The AADs disopyramide, quinidine, and propafenone prolonged APD_90_ and decreased dVdt_max_ and CV in a dose-dependent manner in drug-free versus drug-bound AF conditions. (5) Disopyramide prolonged WL. Disopyramide is more effective against *Pitx2*-induced AF than quinidine and propafenone.

### 3.2. Compared to Other Forms of AF

AF is the most common cardiac arrhythmia with well-established clinical and genetic risk components. AF can be broadly divided into two types, genetic and acquired types, according to the factors that cause AF. Acquired AF usually begins in a self-terminating paroxysmal form (pAF) and this pattern often evolves to become a chronic form (cAF) [1]. In addition to acquired AF, genetic factors are presumed as key in the development of AF, especially in familial cases (fAF) without cardiac pathology. The total genetic contribution to fAF risk can be broadly divided into three components: Rare coding variation, common variation, and undiscovered variation [90].

For common variation, the genetic loci associated with fAF were first identified and are located on chromosome 4q25, upstream of the transcription factor gene *Pitx2* [10]. *Pitx2* deficiency resulted in electrical and structural remodelling, and impaired repair of the heart in murine models [18]. Then, meta-analysis of AF cases identified a novel locus for fAF (*ZFHX3*, rs2106261) [3]. Furthermore, a meta-analysis of multiple well-phenotyped GWASs identified six new susceptibility loci for fAF, including 1q24 (*PRRX1*), 7q3 (*CAV1*), 14q23 (*SYNE2*), 9q22 (*FBP1* and *FBP2*), 15q24 (*HCN4*), and 10q22 (upstream of *SYNPO2L* and *MYOZ1*) [5]. By meta-analyses of SNP associations with AF, researchers discovered five novel loci near the genes *NEURL* (rs12415501 and rs6584555), *GJA1* (rs13216675), *TBX5* (rs10507248), *CAND2* (rs4642101), and *CUX2* (rs6490029)[9]. Five GWASs were conducted in 2017, with a total of no more than 30 novel loci identified. Roselli et al. conducted the largest GWAS meta-analysis, and found that there were 97 loci significantly associated with AF [8]. Nielsen et al. found 142 independent risk variants at 111 loci and prioritised 151 functional candidate genes likely to be involved in fAF. Of these, 80 loci have not been previously reported [91]. Among these common genes, the links between fAF and *Pitx2* or *TBX5* have been deeply investigated. Experimental studies demonstrated that *TBX5* directly activated *Pitx2*, and *TBX5* and *Pitx2* antagonistically regulated the membrane effector genes *SCN5A*, *GJA1*, *RYR2*, *DSP*, and *ATP2A2* [17].

For rare coding variation, *S140G* in the *KCNQ1* gene was the first identified [92]. The potassium voltage-gated channel subfamily E genes encode the regulatory β-subunits of the channels producing the delayed rectifier potassium current. Gain-of-function mutations in *KCNE1*, *KCNE2*, *KCNE3*, and *KCNE5* have been associated with fAF. Furthermore, the genes (i.e., *KCNA5*, *KCND3*, and *KCNH2*) coding the α-subunit of the voltage-gated potassium channels Kv1.5, Kv4.3, and Kv11.1, and the α-subunit of the inwardly rectifying potassium channels Kir2.1, Kir3.4, and Kir6.1 (i.e., *KCNJ2*, *KCNJ5*, and *KCNJ8*) also showed significant associations with fAF risk. fAF-associated potassium channel variants have a gain of channel function, with an expected shortening of the atrial action potential duration and atrial refraction period. The fAF-associated sodium channel genes included *SCN5A*, *SCN10A*, and genes coding sodium voltage-gated channel β subunit 1–4 (*SCN1B*, *SCN2B*, *SCN3B*, and *SCN4B*). The mutations in *SCN5A* exhibited a compromised peak sodium current and an increased sustained sodium current. The related variations in *SCN10A* were implicated in the modulation of the late sodium current, while mutations in *SCN1B-4B* were engaged with the modulation of the inward sodium current. Several non-ion channel genes did not directly alter the action potential, but instead would be expected to instigate the onset of fAF through alternative mechanisms. These included *NPPA*, *NUP155*, *LMNA*, *GJA5*, *AGT*, and *ACE* [93].

For pAF, APD_90_, I_CaL_, and I_Ncx_ were unaltered, indicating the absence of AF-induced electrical remodelling [94]. In contrast, there were increases in SR Ca^2+^ leaks and the incidence of delayed after-depolarisations in pAF. Ca^2+^ transient amplitude and sarcoplasmic reticulum Ca^2+^ load were larger in pAF. Ca^2+^ transient decay was faster in pAF, but the decay of caffeine-induced Ca^2+^ transients was unaltered, suggesting increased SERCA2a function. In agreement, phosphorylation (inactivation) of the SERCA2a inhibitor protein phospholamban was increased in pAF. Ryanodine receptor fractional phosphorylation was unaltered in pAF, whereas ryanodine receptor expression and single-channel open probability were increased. Increased diastolic sarcoplasmic reticulum Ca^2+^ leaks and related delayed after-depolarisations promote cellular arrhythmogenesis in pAF patients. Biochemical, functional, and modelling studies point to a combination of increased sarcoplasmic reticulum Ca^2+^ load related to phospholamban hyperphosphorylation and ryanodine receptor dysregulation as underlying mechanisms.

In addition to Ca^2+^ handling, cAF involves electrophysiological and structural remodelling. Differences in cardiomyocyte Ca^2+^-handling properties between patients with pAF and cAF point towards progressive changes in atrial Ca^2+^ handling due to AF itself [95]. Ca^2+^ transient amplitude is increased in pAF, but decreased in cAF [96]. Consistent with its role as a frequency sensor, CaMKII autophosphorylation and CaMKII-dependent RyR2 phosphorylation are elevated in cAF but not pAF. Rate induced Ca^2+^ loading causes enhanced binding of Ca^2+^ to calmodulin, which activates the phosphatase calcineurin. Calcineurin then dephosphorylates the nuclear factor of activated T cells, which translocates into the nucleus and inhibits the production of *CACNA1C* mRNA, decreasing the message for the I_CaL_ α-subunit, thereby reducing its protein expression and ion transport function [97]. In parallel, the nuclear factor of activated T cells suppresses the production of microRNA (miR)-26 by binding to and negatively regulating sites upstream of the transcriptional start site in the human and mouse atrium [98]. One of the binding targets of the miR-26 seed site is *KCNJ2*, the gene encoding the I_K1_ channel. Reduced miR-26 expression caused by AF removes miR-26-induced destabilisation of the *KCNJ2* message and inhibition of its translation. Inward rectifier current functional expression is enhanced by this mechanism, as well as by increased I_KACh_ in human and canine models [99]. Recent work also shows that NLRP3 activation increases the gene expression of the channels subunits underlying the atrial selective currents I_Kur_ and I_KACh_ [100]. In addition to these pathways, protein kinase isoform switches upregulate the constitutive activity of I_KACh_, and the upregulation of two-pore and Ca^2+^-dependent K^+^ channels [101]. In addition to AF-induced electrical remodelling, rapid cardiomyocyte firing leads to fibroblast activation via a diffusible substance in HL-1 atrial-derived cardiomyocytes, which seems to be ROS-derived from cardiomyocyte nicotinamide adenine dinucleotide phosphate oxidase stimulated by Ca^2+^ loading [102]. ROS are known to activate NLRP3 inflammasomes in other systems and NLRP3 inflammasome activation is known to cause atrial fibrosis [103].

### 3.3. Changes in Human Atrial Electrical Activity Are Linked to Pitx2-Induced Remodelling

Research studies on *Pitx2*-induced AF have demonstrated that *Pitx2*-induced remodelling contributes to atrial cellular electrophysiological changes in AF patients [11,17,26]. Experimental data showed that alterations in human atrial electrical activity included AP shortening, increased upstroke velocity, and decreased conduction velocity [58] and our simulated effects of *Pitx2*-induced remodelling on atrial electrical activity (Table 2 and Table 5) are concordant with these experimental findings.

Previous experiments on isolated human atrial myocytes have demonstrated that *Pitx2*-induced remodelling of ion channels, particularly for I_Ks_ and I_CaL_, may contribute to the clinically significant association between impaired *Pitx2* and AF [26]. Further simulations in our previous study indicated the *Pitx2*-induced changes in I_Ks_ and I_CaL_ led to APD shortening, facilitating sustained re-entry in 3D anatomical atrial geometry [30]. Our simulations in the *Pitx2*-induced AF population of models show that AP shortening is determined by I_CaL_ and I_Ks_, and APD is negatively correlated with I_Ks_ and positively correlated with I_CaL_ (Figure 4). This is consistent with findings from previous studies that the upregulation of I_Ks_ and downregulation of I_CaL_ due to *Pitx2*-induced remodelling critically contribute to the abbreviation of APD [30,33].

Increased upstroke velocity in *Pitx2*-induced AF may result from remodelled I_Na_ arising from impaired *Pitx2*. Previous experiments in atrial cardiomyocytes of *Pitx2*-deleted mice showed increased action potential amplitude and shortened APD [17]. Further gene analysis showed that a reduced *Pitx2* dose caused an increase in the expression of *SCN5A* [17] encoding the alpha subunit of the human cardiac voltage-gated sodium channel (I_Na_). In addition, an increase in the mRNA level of *SCN1B* encoding a beta-1 subunit of voltage-gated sodium channels (I_Na_) was observed in patients with familial AF arising from the *Pitx2c* mutation p.Met207Val [63]. Decreased mRNA levels of *SCN5A* and *SCN1B* were also observed in atrial-specific *Pitx2* mutant mice [22,25], but changes in T-box transcription factor *Tbx5* was not investigated and the downregulation of sodium channel genes may result from impaired *Tbx5*. Previous experiments in atrial cardiomyocytes of *Tbx5*-deleted mice showed reduced *Pitx2* and sodium channel genes, although reduced *Pitx2* without alterations in *Tbx5* caused an increase in sodium channel genes [17]. Based on these data on remodelled I_Na_ due to impaired *Pitx2*, a *Pitx2*-induced AF model was developed and predicted an increase in action potential amplitude and maximum upstroke velocity [36]. Consistent with these findings, our simulations in the *Pitx2*-induced AF population of models show that increased upstroke velocity is determined by I_Na_ and is positively correlated with I_Na_ (Figure 4).

Furthermore, slow conduction in *Pitx2*-induced AF may be associated with reduced gap junctions due to impaired *Pitx2*. Previous experiments in atrial cardiomyocytes of *Pitx2*-deleted mice showed a reduction of *GJA1* encoding the protein connexin 43 (Cx43) [11,17,22,63]. Based on experimental data on remodelled Cx43, our previous simulations showed slow conduction velocity [29,36]. Consistent with these findings, our results in the present study predicted that reduced gap junctions led to the slow conduction of AP propagation.

### 3.4. Antiarrhythmic Effects of Class I Drugs in Pitx2-Induced AF

Despite the prevalence of AF and decades of research, antiarrhythmic therapies for AF continue to have limited efficacy and safety [35]. A population-based study assessed the influence of AF-associated loci on the response to antiarrhythmic drug therapies and showed that carriers of the variant allele at rs10033646 on chromosome 4q25 (*Pitx2*) responded favorably to class I AADs [34]. Class I AADs frequently used clinically include flecainide, disopyramide, quinidine, and propafenone [35]. The antiarrhythmic effects of flecainide on *Pitx2*-induced AF were investigated in our previous study and simulation results indicated that flecainide has antiarrhythmic effects on AF due to impaired *Pitx2* by preventing spontaneous calcium release and increasing wavelength [36]. In the present study using the Bai et al model, the effectiveness of disopyramide, quinidine, and propafenone was assessed with WL. The simulated results showed that disopyramide and quinidine were moderately effective by prolonging WL, whereas propafenone was shown to be ineffective by abbreviating WL in *Pitx2*-induced AF. WL is the product of APD_90_ and CV. Increased APD_90_ and decreased CV occurred upon the application of disopyramide, quinidine, and propafenone. Under the application of disopyramide or quinidine, the extent of AP prolongation is larger than the degree of CV reduction, resulting in an increase in WL. In contrast, the extent of CV reduction is much larger than the degree of AP prolongation, leading to a reduction in WL. Alterations in WL can be linked to blocking effects of class I drugs on ion channels which included G_Na_, G_CaL_, G_to_, G_Ks_, G_Kr_, G_KAch,Ado_, G_KATP_, G_Kur_, and G_K1_. Under the application of class I drugs in *Pitx2*-induced AF, the inhibition of sodium currents would decrease dVdt_max_ and thereby reduce CV, while the blocking of potassium currents would prolong APD_90_. Therefore, the antiarrhythmic effects of class I drugs in *Pitx2*-induced AF can be attributed to their potent blocking of sodium and potassium channels. The effectiveness of drugs is determined by the extent of their blocking effects on sodium and potassium channels. Consistent with findings obtained from the Bai et al. model, the results obtained from the Grandi et al. model show that disopyramide is more effective than propafenone in *Pitx2*-induced AF and this is because the extent of their blocking effects on potassium channels is larger than that on sodium channels. Collectively, our results demonstrate that a combined blocking of sodium and potassium currents can exert synergistic antiarrhythmic effects and therefore is a valuable therapeutic for *Pitx2*-induced AF.

By slowing the rate of atrial flutter/fibrillation, quinidine can decrease the degree of atrioventricular blocking and cause an increase, sometimes marked, in the rate at which supraventricular impulses are successfully conducted by the atrioventricular node, with a resultant paradoxical increase in ventricular rate [104]. The previous study showed that quinidine causes greater QT prolongation in women than in men at equivalent serum concentrations. This difference may contribute to the greater incidence of drug-induced torsades de pointes observed in women taking quinidine and has implications for other cardiac and noncardiac drugs that prolong the QTc interval. The adjustment of dosages based on body size alone is unlikely to substantially reduce the increased risk of torsades de pointes in women [105].

In addition, the direct effect of the class 1a AADs quinidine and disopyramide on APs is significantly modified by their anticholinergic actions. Inhibiting vagal activity can lead to a reduction in I_KAch_ and an increase in sinoatrial rate. Quinidine depressed adenosine-induced I_KAch_, while the effect of disopyramide on the adenosine-induced current was much smaller than that on the Ach-induced current. An experimental study suggested that quinidine may inhibit the muscarinic K^+^ channel itself and/or G proteins, while disopyramide may mainly block functions of muscarinic Ach receptors in atrial myocytes [106]. Although disopyramide may effectively depress atrial rate during flutter, it may lead to an increase in ventricular rate because of an increase in the number of impulses conducted through the atrioventricular node, thereby requiring concomitant treatment with a beta blocker to slow atrioventricular nodal conduction.

### 3.5. Model and AP Shape Independence of Prediction of the Antiarrhythmic Effects of Class I Drugs

It is well known that human atrial myocytes display two distinct AP morphologies: the type-1 AP shows a notch-and-dome morphology and the type-3 AP shows a typical triangular morphology [58,60]. Therefore, the Bai et al. model displaying the type-1 AP [27] and the Grandi et al. model displaying the type-3 AP [59] were chosen to generate populations of human atrial AP. The Bai et al. model was developed by taking into account ionic differences between atrial and ventricular cells based on the previous human ventricular model (TP model) developed by ten Tusscher and Panfilov [107]. The TP model included a subspace calcium variable that controls the dynamics of the I_CaL_ and the ryanodine receptor current. The phenomenological description of I_CaL_-induced calcium release was used with a reduced version of the ryanodine receptor Markov model developed by Stern et al. [108] and Shannon et al. [109]. The AP profile of the Bai et al. model is a spike-and-dome-type action potential and is comparable to the Courtemanche et al. model [110]. The Grandi et al. model was developed largely based on their previous human ventricular model, and therefore the formulation of transmembrane currents differs significantly from the Bai et al. and Courtemanche et al. models. In addition to the main transmembrane currents, the Grandi et al. atrial model also includes the formulation of two chloride currents and a potassium plateau current. The calcium subsystem model is based on the one in the rabbit ventricular model by Shannon et al. [109]. Therefore, the calcium transient of the Grandi et al. model is comparable to the Bai et al. model, but the Grandi et al. model with a triangular AP shape is different from the Bai et al. model. This difference can be reflected by the effects of ion currents on depolarisation and repolarisation. Consistent with this study [111], the excitability of the Bai et al. model is modulated by both I_Na_ and I_K1_, whereas that of the Grandi et al. model is regulated only by I_Na_. During the repolarisation duration, I_to_ and I_Kur_ regulate the AP shape (the notch) at phase 1 and I_Ks_ and I_Kr_ are the important potassium currents that regulate APD_90_. These characteristics are derived from those of the base model (TP model). However, there is no notch shape in the AP of the Grandi et al. model and therefore all potassium currents (including I_to_ and I_Kur_) are modulated APD_90_. Among these potassium currents, I_Ks_, I_KAch_, and I_KATP_ have small effects on APD_90_.

According to the stimulation protocol (1Hz) used in experiments [58], we ran simulations and reproduced Pitx2-induced AP morphology, with both dVdt_max_ increases and AP shortening being reported experimentally [58]. Further simulations predicted the antiarrhythmic effects of class I drugs on APD_90_ and dVdt_max_, but their effects are dependent on the baseline AP morphology [112,113] seen in previous modelling [54]. While some of the differences in model responses might be related to distinct AP morphologies, we cannot exclude model dependencies due to distinct cellular model structure and models of ionic and calcium handling processes. Interestingly, analyses using both models demonstrated that the class I drugs disopyramide, quinidine, and propafenone prolonged APD_90_ and reduced dVdt_max_ in a dose-dependent manner. Therefore, these results demonstrated that these class I drugs consistently produced anti-AF effects independent of the baseline electrophysiological characteristics.

Although the class I drugs disopyramide, quinidine, and propafenone produced similar anti-AF effects (AP prolongation and dVdt_max_ and CV reduction), the degree of changes in APD_90_ and dVdt_max_ is different between the Bai et al. model and the Grandi et al. model. According to the extents of dVdt_max_ and CV reductions, drugs in both Bai et al. and Grandi et al. models are sorted as propafenone>quinidine>disopyramide. However, based on the degrees of AP prolongation, the drug ranking obtained with the Bai et al. model is quinidine>disopyramide>propafenone, whereas drugs whose effects were predicted with the Grandi et al. model are ranked as disopyramide>quinidine>propafenone. Different alterations in CV and APD_90_ lead to different changes in WL, which can be used as one of the indexes for evaluating the antiarrhythmic effects of drugs at the tissue level [54]. If the extent of AP prolongation is larger than that of CV reduction, WL is prolonged. Conversely, WL is shortened. For the Bai et al. model, WL is prolonged upon the application of quinidine and disopyramide, but is shortened upon the application of propafenone. However, for the Grandi et al. model, WL upon the application of disopyramide at various concentrations, quinidine at the low and medium concentrations, and propafenone at the low concentration is prolonged, but is shortened upon the application of quinidine at the high concentration and propafenone at the medium and high concentrations. Based on alterations in WL, the drug ranking obtained with the Bai et al. model is quinidine>disopyramide>propafenone, whereas drugs are sorted as disopyramide>quinidine>propafenone for the Grandi et al. model. Therefore, analyses using both models demonstrated that quinidine and disopyramide are more effective against *Pitx2*-induced AF than propafenone.

### 3.6. Limitations and Future Work

Several limitations specific to this study are addressed here. Firstly, the electrophysiological representation of AF-induced remodelling in the human atrial AP model is based on data from previous mouse models of *Pitx2*-induced AF because of the lack of experimental data on humans. Although shortened atrial repolarisation, a more depolarised resting membrane potential, and abnormalities in calcium cycling were observed in atrial cardiomyocytes of mice with reduced *Pitx2* mRNA and these models (including Bai et al. and Grandi et al. models) could reproduce these phenomena in our previous study [27], *Pitx2*-induced AP data of human atrial cardiomyocytes and 3D whole atria for reproducing P waves [12] are not available to further validate the suitability and accuracy of our models to date. Special attention must be paid to the differences between mouse and human atrial cells [114]. In general, APD is approximately 50 ms in mice, compared to 250 ms in humans. The AP morphology reflects the contribution of numerous depolarising and repolarising currents. Even when the same type of ion channel is expressed in humans and mice, its contribution to the AP morphology may differ substantially, given the large difference in APD. In the mouse heart, I_CaL_ contributes less to the AP than in humans and therefore the murine AP shows a gradual repolarisation rather than a distinct plateau phase. The much faster repolarisation in mice is mediated by transient outward K^+^ currents with a fast and slow recovery from inactivation, a slowly inactivating K^+^ current, and a non-inactivating, steady-state current. In humans, the transient outward K^+^ current with a fast recovery is mainly involved in phase 1 repolarisation, with more prominent expression in the epicardium. In addition, the rapid and slow delayed outward rectifier K^+^ currents are predominantly responsible for phase 3 repolarisation. However, studies in mice did observe the rapid and slow delayed rectifier K+ currents, but their contribution to repolarisation under physiological conditions is probably negligible or minor [115]. Further, we simulated AF model populations with all identified targets associated with impaired *Pitx2* [11,17,18,22,23,26]. However, *Pitx2*-induced remodelling is different in published experimental studies [11,17,18,22,23,26], atrial cell types [11], AF stages [116], and AF patients of different ages [117]. The complexity of Pitx2-induced remodelling may present complex responses to these class I drugs and require more realistic heterogenous descriptions in cellular and tissue simulations. Secondly, we assumed that these changes in mRNA expression are quantitatively reflected at the final functional level of ion channels to obtain human AF myocyte models that reproduced the experimentally observed changes in the mRNA levels corresponding to key proteins under *Pitx2*-induced electrical remodelling conditions. Of note, mRNA alterations often do not match electrophysiological alterations and this is one of the main limitations. Thirdly, the effects of these class I AADs on ion currents were modelled by changing maximum current conductances with a simple pore blocking scheme based on IC_50_ and nH values, but it might be important to incorporate the state dependence and use dependence of antiarrhythmic drugs in evaluating realistic compounds [37,61,89]. Model variations for these characteristics (including the affinity of the drug compound to various gating modalities, binding kinetics, drug polarity, charge for the drug-channel interaction, and so on) are far beyond the scope of the current study and should be explored in future studies. In addition, IC_50_ values were chosen based on experimental data from atrial cells (where data are available) and large ventricular cells (where atrial data are not available), but the large variability in IC_50_ was influenced by various experimental conditions, including different species, different cell types [118], temperature [119], ionic concentrations, and voltage clamp protocols. Therefore, special attention should be paid to explain our simulated results. Fourthly, considering the stimulation protocol used under the experimental conditions [58], we performed all simulations only at 1Hz and rate-dependent APD modulation will be further investigated when experimental data at different frequencies are available. Here, we only used APD_50_, APD_90_, RMP, and dVdt_max_ as biomarkers to calibrate populations of human atrial AP models. Parameter unidentifiability, which can be attenuated if broader experimental datasets are available, is also a potential limitation. Fifthly, we analysed the effects of class I drugs by quantifying changes in APD_90_ and dVdt_max_ at the cellular level and CV and WL in tissue. This approach is consistent with the accepted mechanisms of action of class I drugs and is limited in that it did not account for composite metrics, including calcium-associated biomarkers, after-depolarisations, alternans, and AP propagation dynamics. Sixthly, [*Ach*]_0_ was set to 0 nM, but the contribution of G_KAch,Ado_ to changes in AP could be under-estimated. Indeed, the “extracellular” concentration of adenosine and therefore G_KAch,Ado_ could be variable [120]. The adenosine receptor is a Gi protein-coupled receptor and is predominantly expressed in atrial and nodal tissue. In the atrial myocardium, adenosine shortens both APD and causes RMP hyperpolarisation, thereby increasing the fraction of available sodium channels [121]. Such an increased sodium current, combined with shorter refractory periods, would be expected to increase the vulnerability and sustainability of AF. This action is mediated by the I_KAch_ channel effector system under basal conditions. With increased adrenergic stimulation, adenosine attenuates the catecholamine-stimulated contractility and cyclic AMP accumulation. Adenosine attenuates both catecholamine-activated outward potassium and inward calcium currents, presumably by the inhibition of the phosphorylation of these channels, because adenosine inhibits adenylate cyclase activity and decreases the generation of cyclic AMP. The net electrophysiologic effect is the result of the direct adenosine-mediated activation of I_KAch_ channels and the indirect adenosine-mediated inhibition of potassium and calcium currents. In the sinoatrial node, adenosine slows the heart rate. The pacemaker slowing caused by adenosine is due to a hyperpolarisation of the membrane which results in a decrease in phase 4 depolarisation. The hyperpolarisation is partly accomplished by the activation of I_KAch_ channels through the Gi βγ-subunits and partly by the reduction in cyclic adenosine monophosphate (cAMP) through the Giα-subunit. Reduced cAMP levels decrease hyperpolarisation-activated cyclic nucleotide-gated (HCN) channel activity in pacemaker cells and decrease protein kinase A activity, thereby reducing I_CaL_ [122]. These factors were not considered in the present study and special attention should be paid to explain our simulated results. Finally, further studies using populations of 3D atrial models should be conducted to show whether class I AADs really inhibit *Pitx2*-induced AF. However, whole heart simulations with complex models and large parameter dimensions are computationally expensive and generally require high-performance computing resources; running millions (or more) of simulations to obtain estimates of model outputs is likely to be prohibitively expensive. Further studies will be performed if conditions permit. Nevertheless, mechanisms underlying the actions of class I drugs are highly complex and future investigations should be carried out.

## 4. Materials and Methods

### 4.1. Experimental Dataset

The experimental dataset of AP recordings was used in the present study for calibration of human atrial electrophysiology. The dataset published in the previous study [58] comprised 480 instances from 379 patients: AP recordings from *n* = 256 right atrial appendages of N = 221 SR patients and from *n* = 224 right atrial appendages of N = 158 AF patients. Human myocytes were isolated enzymatically from atrial appendages and APs were recorded with standard intracellular microelectrodes in atrial trabeculae. Preparations were electrically stimulated at a single constant rate of 1 Hz for 60 min with isolated square-wave stimuli of a 1 ms duration, two times the threshold intensity. Obtained human atrial APs displayed spike-and-dome and more triangular conformations. The following parameters were quantified to characterise inter-subject variability in human atrial AP: dVdt_max_, RMP, and APD_50_ and APD_90_. Compared with the APs of SR myocytes, APD_50_ and APD_90_ were reduced and dVdt_max_ was increased for the APs of AF atrial cells (Table 2). More information (including ethics approval, informed consent, and basic information of participants) regarding the experimental conditions under which the data were collected is available in the study of Ravens et al. [58].

### 4.2. Mathematical Models Representing Distinct Morphological Subtypes of Human Atrial AP

To represent the spike-and-dome and triangular conformations of human atrial APs observed in experiments [58,60], the Bai et al. model displaying a type-1 AP with notch-and-dome morphology and the Grandi et al. model displaying a type-3 AP with typical triangular shape were chosen as a base to construct the computational AP model populations.

The Bai et al. model developed by our group is able to reproduce human AP morphology, APD rate dependence, and triggered activity, i.e., early after-depolarisations (EADs), delayed after-depolarisations (DADs), and spontaneous depolarisations [27]. This biophysically detailed model of human atrial cellular electrophysiology was also used to investigate mechanisms underlying Pitx2-induced AF and here we provide a brief description. It includes representation of the 13 transmembrane ionic currents and 2 main intracellular calcium flows, including fast sodium current (I_Na_), transient outward potassium current (I_to_), rapid delayed rectifier potassium current (I_Kr_), slow delayed rectifier potassium current (I_Ks_), ultrarapid delayed rectifier potassium current (I_Kur_), inward rectifier potassium current (I_K1_), L-type calcium current (I_CaL_), background sodium current (I_BNa_), background calcium current (I_BCa_), plateau potassium current (I_PK_), plateau calcium current (I_PCa_), sodium–potassium pump current (I_NaK_), sodium–calcium exchange current (I_Ncx_), the calcium flow (I_up_) through the sarcoplasmic reticulum calcium ATPase (SERCA), and calcium release flow (I_rel_).

Comparative simulations were carried out using the human atrial cell model of Grandi et al. [59,123] (GB model). The baseline GB model was modified to generate the Grandi et al. model to facilitate AP propagation in tissue by replacing the I_Na_ formulation with that of the human cell model [107,124]. The Grandi et al. model includes representation of the 18 transmembrane ionic currents and 2 main intracellular calcium flows, including I_to_, I_Kr_, I_Ks_, I_Kur_, I_K1_, I_CaL_, I_BNa_, I_BCa_, I_PK_, I_PCa_, I_NaK_, I_Ncx_, I_up_, I_rel_, sodium current through the L-type calcium channel (I_CaNa_), potassium current through the L-type calcium channel (I_CaK_), calcium-activated chloride current (I_ClCa_), background chloride current (I_BCl_), late sodium current (I_NaL_), and I_Na_. This I_Na_ model [107,124] is given by
(1)$$I_{Na}=G_{Na}m^{3}hj\left(V_{m}-E_{Na}\right)$$
(2)$$m_{\infty }=1.0/{\left[1.0+e^{\langle -56.86-V_{m}\rangle /9.03}\right]}^{2}$$
(3)$$\tau_{m}=\left\{1.0/\left[1.0+e^{\langle -60-V_{m}\rangle /5}\right]\right\}{\left\{0.1/\left[1.0+e^{\langle V_{m}+35\rangle /5}\right]+0.1/\left[1.0+e^{\langle V_{m}-50\rangle /200}\right]\right\}}_{\;}$$
(4)$$h_{\infty }=1.0/{\left[1.0+e^{\langle 71.55+V_{m}\rangle /7.43}\right]}^{2}$$
(5)$$\alpha_{h}=0\;\left(V_{m}\geq -40\right)\;\alpha_{h}=0.057e^{-\langle V_{m}+80\rangle /6.8}\;(V_{m}<-40)$$
(6)$$\beta_{h}=0.77/\left[0.13\left(1.0+e^{-\langle 10.66+V_{m}\rangle /11.1}\right)\right]\;\left(V_{m}\geq -40\right)\;\beta_{h}=2.7e^{0.079V_{m}}+310000e^{0.3485V_{m}}(V_{m}<-40)$$
(7)$$\tau_{h}=1.0/\alpha_{h}+\beta_{h}$$
(8)$$j_{\infty }=1.0/{\left[1.0+e^{\langle 71.55+V_{m}\rangle /7.43}\right]}^{2}$$
(9)$$\alpha_{j}=0\;\;\;\;\;\;\;\;\;\;\;\;\;\;\;\;\;\;\;\;\;\;\;\;\;\;\;\;\;\;\;\;\;\;\;\;\;\;\;\;\;\;\;\;\;\;\;\;\;\;\;\;\;\;\;\;\;\;\;\;\;\;\;\;\;\;\;\;\;\;\;\;\;\;\;\;\;\;\;\;\;\;\;\;\;\;\;\;\;\;\;\;\;\;\;\;\;\left(V_{m}\geq -40\right)\;\alpha_{j}=\langle -25428e^{-(80+V_{m)/6.8}}+0.000006948e^{-0.04391V_{m}}\rangle \left(37.78+V_{m}\right)/\left(1.0+e^{0.311\langle 79.23+V_{m}\rangle }\right)\;(V_{m}<-40)$$
(10)$$\begin{array}{ll} \beta_{j}=\frac{0.6e^{0.057V_{m}}}{1.0+e^{-0.1\langle 32+V_{m}\rangle }} & \left(V_{m}\geq -40\right)\\ \beta_{j}=0.02424e^{-0.01052V_{m}}/\left(1.0+e^{-0.1378\langle 40.14+V_{m}\rangle }\right) & \left(V_{m}<-40\right) \end{array}$$
(11)$$\tau_{j}=1.0/\alpha_{j}+\beta_{j}$$
where G_Na_ (14.838 nS/pF) is the maximal conductance, m, h and j are three gate variables for I_Na_, V_m_ is the membrane potential, and E_Na_ is the sodium equilibrium potential. m∞, h∞, and j∞ denote steady-state activation, steady-state inactivation, and steady-state inactivation, respectively.τ_m_, τ_h_, and τ_j_ are the time constants for m∞, h∞, and j∞, respectively.

In additional to the original ion currents of the Bai et al. and Grandi et al. models, I_KATP_ and I_KAch_ were incorporated into these models. I_KAch_ channel activation shortens both APD and the effective refractory period causes RMP hyperpolarisation, thereby increasing the fraction of available sodium channels. Such an increased sodium current, combined with shorter refractory periods, would be expected to increase the vulnerability and sustainability of AF. The effects of ATP are broadly similar to those of adenosine and activated I_KAch_ channels will shorten the action potential and reduce the effective refractory period, a significant risk factor for the development of re-entrant arrhythmias like AF. This I_KATP_ model [125] is given by
(12)$$I_{KATP}=G_{KATP}(1.0/(1.0+{\left(\frac{{\left[ATP\right]}_{i}}{K_{ATP}}\right)}^{2})){\left(\frac{{\left[K^{+}\right]}_{O}}{K_{ATP}}\right)}^{0.24}\left(V_{m}-E_{K}\right)$$
where G_KATP_ is the maximal conductance, [ATP]i (6.8 mM for a normal value) is the intracellular nucleotide level, [K^+^]_0_ is extracellular potassium concentration, and E_K_ is the sodium equilibrium potential. The I_KAch_ model [126,127] is given by
(13)$$I_{KAch}=G_{KAch,Ado}r\left(V_{m}-E_{K}\right)/\left(1.0+e^{\left(20+Vm\right)/20}\right)$$
(14)$$\alpha_{r}=0.01232/\left(1.0+0.0042/{\left[Ach\right]}_{0}\right)+0.0002475$$
(15)$$\beta_{r}=0.01e^{0.0133\left(40+Vm\right)}$$
(16)$$r_{\infty }=\alpha_{r}/\alpha_{r}+\beta_{r}$$
(17)$$\tau_{r}=1.0/\alpha_{r}+\beta_{r}$$
where $G_{KAch,Ado}$ (0.135 nS/pF) is the maximal conductance, ${\left[Ach\right]}_{O}$ (0.0 nM for a normal value) is the extracellular Ach level, and r∞ and τ_r_, respectively, denote the steady-state activation and the time constant. This model was developed based on these experimental data on I_Ach_ at the holding potential of −40 mV in the presence of different Ach concentrations [128].

Based on these models of ion channels, the electrophysiological behavior of human atrial cells can be modelled with the following differential equation:(18)$$dV_{m}/dt=-\left(I_{ion}+I_{stim}\right)/C_{m}$$

In order to investigate the electrophysiological behavior of human atrial tissues, a multicellular model of homogeneous atrial tissue with 200 nodes spaced evenly by 0.15 mm was constructed to quantify APD_90_ and CV for calculating WL. The propagation of APs was governed by the partial differential equation:(19)$$dV_{m}/dt=-\left(I_{ion}+I_{stim}\right)/C_{m}+D\left(\partial^{2}V_{m}/\partial x^{2}\right)$$
where D is a scalar coefficient describing the intercellular electrical coupling via gap junctions. In simulations, D was set to be a constant value of 0.154 mm^2^ms^−1^ that gave a CV of 0.65 m/s for the Bai et al. model and 0.59 m/s for the Grandi et al. model. The resulting CVs were comparable to those seen in atrial tissue [129,130]. The time (t) step was 0.02 ms for the Bai et al. model and 0.005 ms for the Grandi et al. model.

### 4.3. Modelling Pitx2-Induced AF

To obtain human AF myocyte models that reproduced the experimentally observed changes in the mRNA levels corresponding to key proteins under Pitx2-induced electrical remodelling conditions, we assumed that these changes in mRNA expression are quantitatively reflected at the final functional level of ion channels [29] and incorporated alterations in the maximal conductances of ionic currents due to Pitx2-induced electrical remodelling into the Bai et al. (or the Grandi et al.) model. These Pitx2-induced changes in the ionic channel properties have been well characterised in many experimental studies [17,18,23,24,25,26,63]. However, the identified targets were different among these studies: While several studies showed no Pitx2-associated changes [18] and reductions [22,25] in sodium channel gene expression, significant increases [17,63] in this channel gene expression have also been well documented. Similarly, calcium channel gene expression was found to be decreased in this study [12], but increased in others [22,23,25,26]. Further, whereas the overexpression of the potassium gene encoding I_K1_ was identified in the study of Kirchhof et al. [11], the underexpression of this gene was observed in some studies [18,22]. In addition, all these studies have shown the upregulation of genes encoding I_Ks_ [11,12,15,26,63], I_rel_ [12,17,23,25], and I_up_ [22,24,66,69,70]. Modifications made for each model are summarised in Table 3.

### 4.4. Simulations of Actions of Class I AADs in Pitx2-Induced AF

In our previous studies [37,89], the actions of the class I AADs disopyramide, quinidine, and propafenone on human atrial electrophysiology were simulated in the setting of hERG-linked short QT syndrome [37]. The effects of these class I AADs on ion currents were modelled by changing maximum current conductances with a simple pore blocking scheme based on IC_50_ and nH [31]. Maximum current conductances associated with these drugs included G_Na_, G_CaL_, G_to_, G_Ks_, G_Kr_, G_Kur_, and G_K1_. For disopyramide, as measured in previous studies, values of IC_50_ for G_Na_, G_CaL_, G_to_, G_Ks_, G_Kr_, and G_Kur_, respectively, were taken to be 168.4 [64], 1036.7 [64], 20.9 [68], 88.1 [71], 14.4 [64], and 25.0 μM [74]. For quinidine, as observed in experimental studies, G_Na_, G_CaL_, G_to_, G_Ks_, G_Kr_, G_Kur_, and G_K1_ were decreased with an IC_50_ of 14.6 [64], 14.9 [66], 21.8 [69], 44.0
[72], 0.72 [64], 6.6 [69], and 42.6 μM [69], respectively. Propafenone decreased G_Na_, G_CaL_, G_to_, G_Kr_, G_Kur_, and G_K1_ with an IC_50_ of 1.2 [65], 1.7 [67], 4.8
[70], 2.0 [73], 4.4 [75], and 16.8 μM [76], respectively. Taking into account plasma protein binging, estimates of the most likely unbound concentrations of disopyramide, quinidine, and propafenone are given as 1–15 [88,131], 1–15 [88,131], and 0.15–1 μM [85,86,87]. The effects of low (L), medium (M), and high (H) doses within the therapeutic ranges of disopyramide (Diso_L: 1.0, Diso_M: 2.0, and Diso_H: 5.0 μM), quinidine (Quin_L: 1.0, Quin_M: 2.0, and Quin_H: 5.0 μM), and propafenone (Prop_L: 0.2, Prop_M: 0.5, and Prop_H: 0.8 μM) were studied.

### 4.5. Generation and Calibration of Populations of Models

To capture inter-subject variability, populations of sampled models of human atrial electrophysiology for SR were generated based on the Bai et al. and the Grandi et al. models. All models in each population shared the same equations but parameters of ionic current conductances in determining the human atrial AP were varied with respect to their original values. These parameters were independently varied following a log-normal distribution and sigma was set to be 0.2 to cover a range of variability similar to that seen in experiments of previous studies [41,54,61,62]. The size of the SR population was set to be 1200 for convergence of the sensitivity coefficients [132]. Following the ASME V&V40 Standard proposed by the Subcommittee of the American Society of Mechanical Engineers (ASME) on Verification and Validation (V&V) in Computational Modeling of Medical Devices [133] for developing a structured approach for establishing the credibility of computational models for a specific use, we used the candidate models to simulate human atrial AP by considering stimulation frequency (1Hz) under the experimental conditions [58] and calculated AP biomarkers of each candidate model. The candidate models generated in the previous step were selected to constitute the SR population whose simulated electrophysiological properties are within 3 standard errors from the mean experimental values in SR patient data on AP biomarkers (including dVdt_max_, RMP, APD_50_, and APD_90_) in Table 2. This step yields the experimentally calibrated population of models. Modifications to ionic currents due to Pitx2-induced remodelling (Table 3) were introduced into SR model variants to generate the candidate AF models. These candidate AF models were calibrated to 3 standard errors from the mean experimental values in AF patient data (Table 2). Then, actions of class I AADs on ion channels were incorporated into the experimentally calibrated AF models to investigate their effects on dVdt_max_, APD_90_, CV, and WL.

### 4.6. Simulation Protocols

The stimulation protocol mimics the one used by Ravens et al. [58] to obtain AP measurements in SR and AF cardiomyocytes. Cell mathematical models were initially preconditioned by pacing at a basic cycle length of 1000 ms until the steady state was reached. A stimulus with an amplitude of −80 μA/μF and a duration of 0.5 ms was applied at each pace.

Considering the computational cost of reaching the steady state, a series of 10 conditioning waves was initiated by supra-threshold stimuli with an amplitude of −80 μA/μF and a duration of 1.0 ms to the 3 nodes at the strand end. The last beat was recorded for APD_90_ and CV analysis.

### 4.7. Sensitivity Analysis

To quantify the relative importance of ionic conductances in determining changes in AP biomarkers, PCCs were used on the SR and AF populations to evaluate the role of each ionic current [132]. Partial correlation is a method to find correlations between two variables, after accounting for the linear effects of one or more additional variables [134]. The PCC between x and y, given the set of N additional variables $z_{i}$, is then defined as the correlation coefficient between the residuals $r_{x}=x-\hat{x}$ and $r_{y}=y-\hat{y}$ [132]. $\hat{x}$ and $\hat{y}$ are the respective sample means or the following linear regression models:(20)$$\hat{x}=c_{0}+\overset{N}{\underset{i=1}{\sum }}c_{i}z_{i}\;\;\;and\;\;\;\hat{y}=b_{0}+\overset{N}{\underset{i=1}{\sum }}b_{i}z_{i}$$
(21)$$PPC\left(x,y,z_{i}\right)=\frac{Cov\left(r_{x},r_{y}\right)}{Var\left(r_{x}\right)Var\left(r_{y}\right)}$$
where *Cov*(*r_x_*,*r_y_*) represents the covariance between *r_x_* and *r_y_*, while $Var\left(r_{x}\right)$ and $Var\left(r_{y}\right)$ are, respectively, the variance of $r_{x}$ and variance of $r_{y}$.

### 4.8. Software, Numerical Methods, and Statistical Analysis

The Bai et al. (available from the repository CellML http://models.cellml.org/workspace/520) and the Grandi et al. models (freely available at https://github.com/drgrandilab) were implemented in MATLAB 2018a (The MathWorks, Natick, MA, USA) using the stiff ordinary differential equation solver ode15s and the analysis of biomarkers was also performed using MATLAB. In addition, the 1D models were implemented in C++. The ordinary differential equations (ODEs) were solved using the forward Euler method. Our user project containing newly created datasets and the simulation codes used in this study is available to download from the GitHub website (https://github.com/aspirerabbit). All simulations and data analyses were performed on a computing cluster with Intel(R) Xeon(R) CPU E5-2690 v4 @ 2.60GHz 32 CPUs 28 CPUs (56 threads) + 128GB. The statistical significance of differences in ionic conductance distributions between populations was evaluated by using the Mann–Whitney U test. A probability <0.05 was considered statistically significant.

## 5. Conclusions

In conclusion, populations of models reproduce the variability in human atrial AP properties measured in samples obtained from patients and AF models predict AP shortening and slow conduction in Pitx2-induced remodelling conditions observed in experiments. State-of-the-art Quantitative Systems Pharmacology Framework simulations demonstrated that disopyramide, quinidine, and propafenone produce AP prolongation and slow conduction in the setting of Pitx2-induced AF. However, disopyramide was more effective in prolonging WL than propafenone and quinidine.

## Figures and Tables

**Figure 1 ijms-22-01265-f001:**
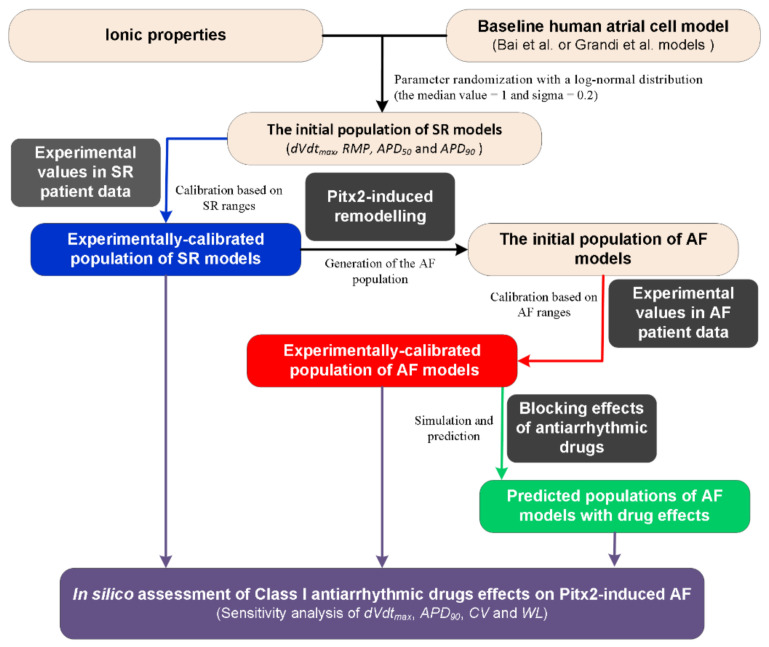
A flow chart illustrating the process for in silico assessment of class I antiarrhythmic drug effects on *Pitx2*-induced atrial fibrillation (AF). Note: There are two distinct morphological subtypes of human atrial action potential (AP) [58] and therefore the Bai et al. model with a notch-and-dome AP morphology [27] and the Grandi et al. model with a triangular AP shape [59] were used in the present study. Class I antiarrhythmic drugs assessed include quinidine, disopyramide, and propafenone. Abbreviations: *Pitx2*: Homeodomain transcription factor 2, SR: Sinus rhythm; dVdt_max_: Maximum upstroke velocity, RMP: Resting membrane potential, APD_50_ and APD_90_: AP duration at 50% and 90%, respectively, CV: Conduction velocity, and WL: Wavelength.

**Figure 2 ijms-22-01265-f002:**
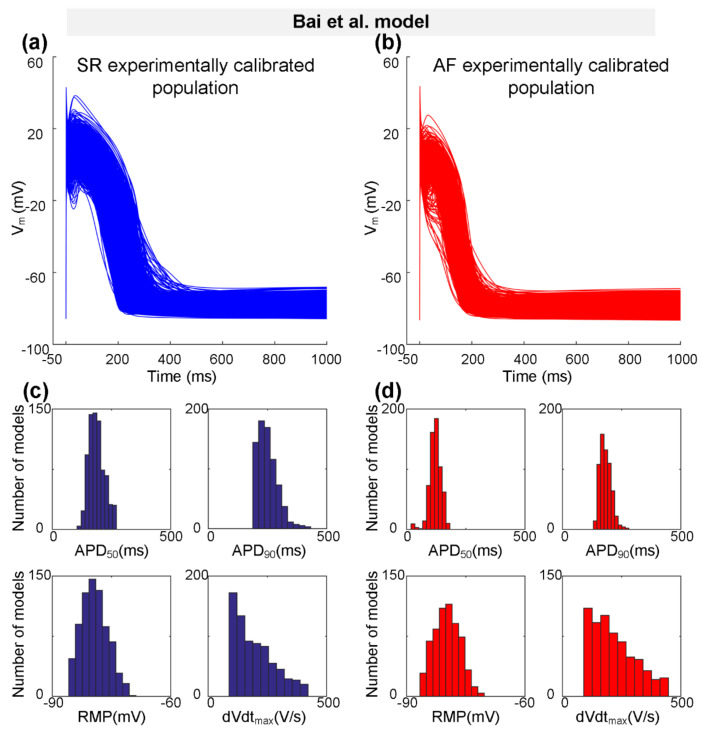
The experimentally calibrated populations constructed with the Bai et al. model for sinus rhythm (SR, blue) and atrial fibrillation (AF, red) conditions. (**a**,**b**) Representative traces of action potential (AP) models in SR and *Pitx2*-induced AF conditions. (**c**,**d**) Distributions of AP biomarkers (including dVdt_max_, RMP, APD_50_, and APD_90_) under SR and *Pitx2*-induced AF conditions.

**Figure 3 ijms-22-01265-f003:**
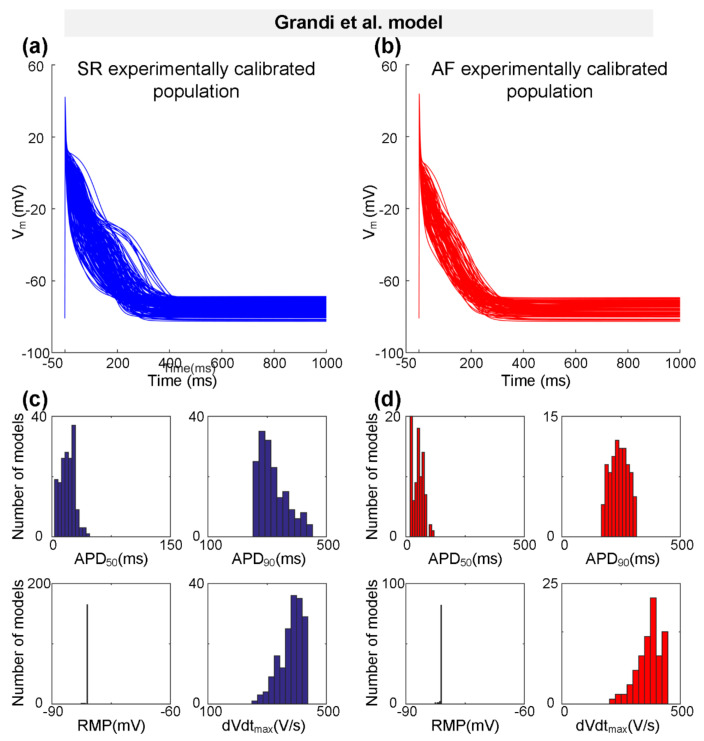
The experimentally calibrated populations constructed with the Grandi et al. model for sinus rhythm (SR, blue) and atrial fibrillation (AF, red) conditions. (**a**,**b**) Representative traces of action potential (AP) models in SR and *Pitx2*-induced AF conditions. (**c**,**d**) Distributions of AP biomarkers (including dVdt_max_, RMP, APD_50_, and APD_90_) under SR and *Pitx2*-induced AF conditions.

**Figure 4 ijms-22-01265-f004:**
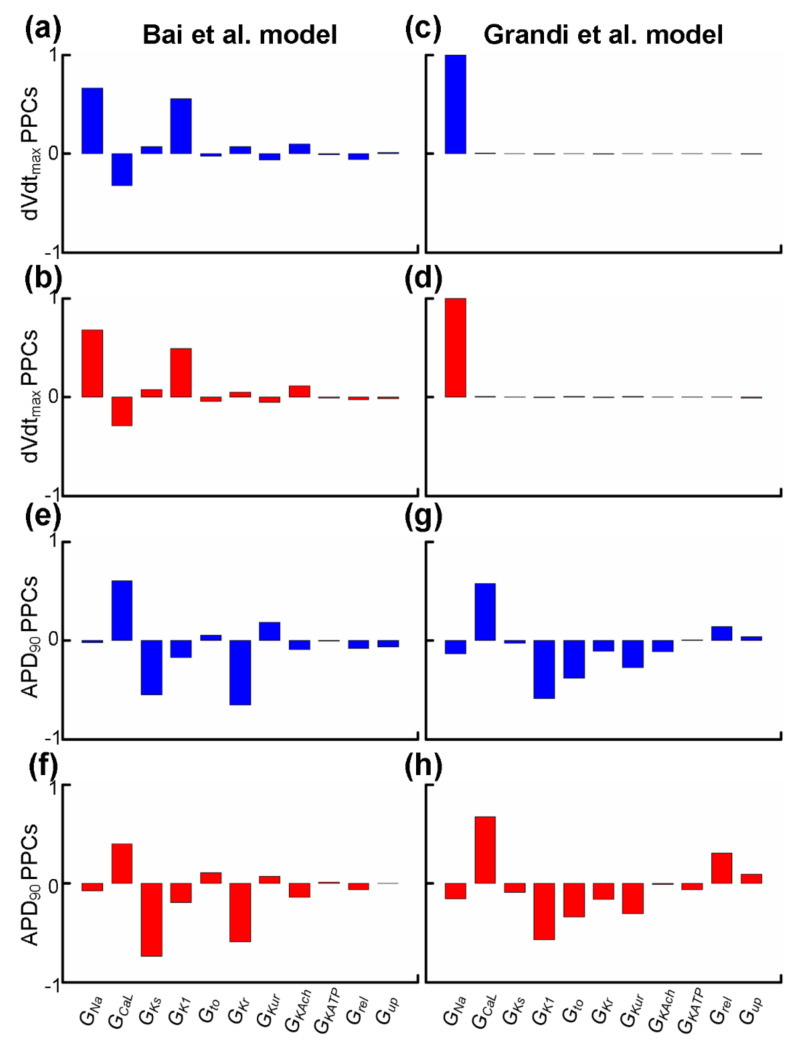
Partial correlation coefficients (PPCs) between biomarkers (dVdt_max_ or APD_90_) and parameters associated with *Pitx2*-induced electrical remodelling (G_Na_, G_CaL_, G_Ks_, G_K1_, G_rel_, and G_up_) or actions (G_Na_, G_CaL_, G_to_, G_Ks_, G_Kr_, G_Kur_, G_KATP_, G_KAch,Ado_, and G_K1_) of antiarrhythmic drugs. PPCs between dVdt_max_ and G_Na_, G_CaL_, G_Ks_, G_K1_, G_Kr_, G_to_, G_Kur_, G_KATP,_ G_KAch,Ado_, G_rel_, or G_up_ for the virtual atrial myocytes created by the Bai et al. model in SR (**a**, blue) and AF (**b**, red) are compared to those for virtual atrial myocytes in the SR (**c**, blue) and AF (**d**, red) populations created by the Grandi et al. model. PPCs of APD_90_ for virtual atrial SR (**e**, blue) and AF (**f**, red) myocyte populations created by the Bai et al. model are compared to those for the virtual atrial myocytes created by the Grandi et al. model in SR (**g**, blue) and AF (**h**, red).

**Figure 5 ijms-22-01265-f005:**
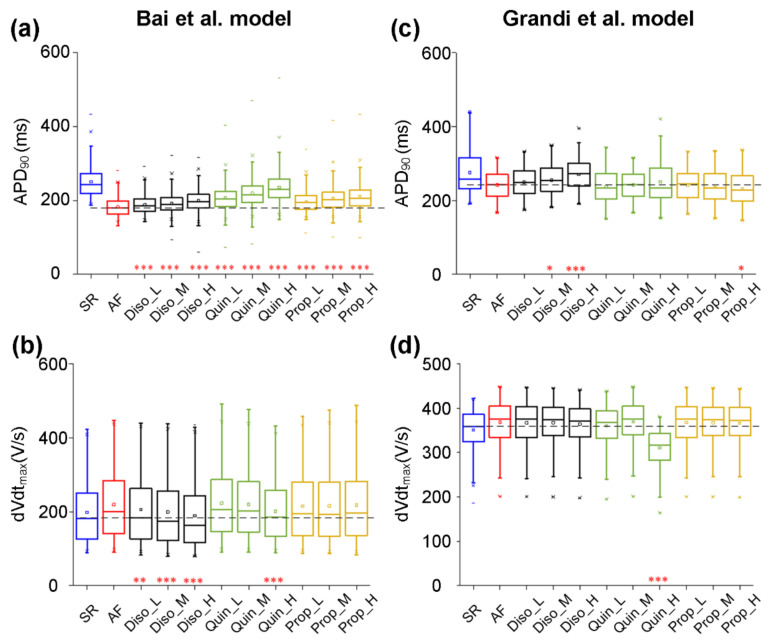
Effects of low (L), medium (M), and high (H) doses of disopyramide (Diso), quinidine (Quin), and propafenone (Prop) on the atrial fibrillation (AF) population. Using the Bai et al. model, simulated changes in APD90 (**a**) and dVdtmax (**b**) following the application of drugs at different doses in comparison to the drug-free conditions in the AF population or in the normal population for sinus rhythm (SR). Using the Grandi et al. model, ranges of APD90 (**c**) and dVdtmax (**d**) in conditions of drug-free SR, drug-free AF, AF in the presence of disopyramide at low (Diso_L), medium (Diso_M), and high (Diso_H) doses, AF in the presence of quinidine at low (Quin_L), medium (Quin_M), and high (Quin_H) doses, and AF in the presence of propafenone at low (Prop_L), medium (Prop_M), and high (Prop_H) doses. Each boxplot represents the range covered by the ionic conductances: The edges of the box are the 1st and 3rd quartiles, the whiskers extend to the most extreme datapoints, the estimated median physiological value is the central horizontal line and the notch around the median is the 5% significance level (Mann–Whitney U test: * *p* < 0.05; ** *p* < 0.01; *** *p* < 0.001).

**Figure 6 ijms-22-01265-f006:**
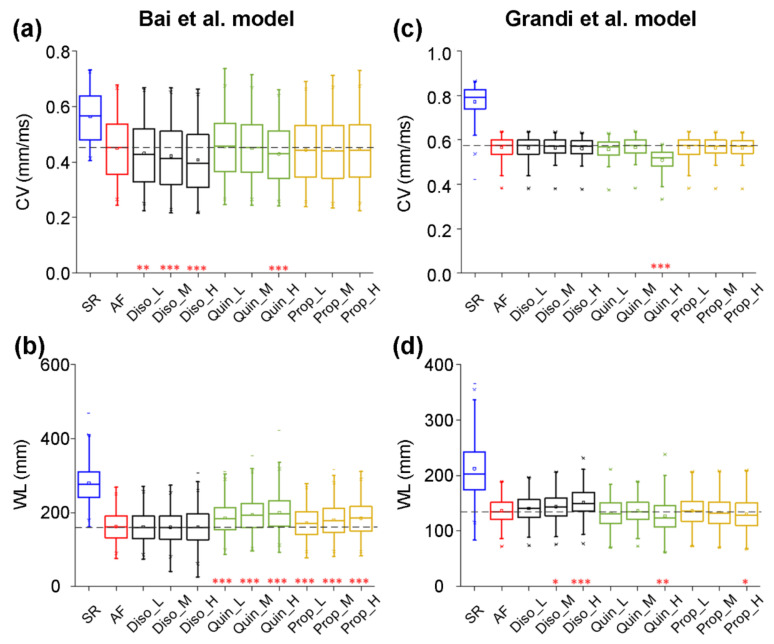
Effects of low (L), medium (M), and high (H) doses of disopyramide (Diso), quinidine (Quin), and propafenone (Prop) on the virtual tissue population for atrial fibrillation (AF). Using the Bai et al. model, simulated changes in conduction velocity (CV, **a**) and wavelength (WL, **b**) following the applications of drugs at different doses in comparison to the drug-free conditions in the AF population or in the normal population for sinus rhythm (SR). Using the Grandi et al. model, ranges of CV (**c**) and WL (**d**) in conditions of drug-free SR, drug-free AF, AF in the presence of disopyramide at low (Diso_L), medium (Diso_M), and high (Diso_H) doses, AF in the presence of quinidine at low (Quin_L), medium (Quin_M), and high (Quin_H) doses, and AF in the presence of propafenone at low (Prop_L), medium (Prop_M), and high (Prop_H) doses. Each boxplot represents the range covered by the ionic conductances: The edges of the box are the 1st and 3rd quartiles, the whiskers extend to the most extreme datapoints, the estimated median physiological value is the central horizontal line and the notch around the median is the 5% significance level (Mann–Whitney U test: * *p* < 0.05; ** *p* < 0.01; *** *p* < 0.001).

**Table 1 ijms-22-01265-t001:** Parameters associated with ionic properties for constructing populations of atrial models.

Parameters	Definition	Models	
		Bai et al. [27]	Grandi et al. [59]
G_Na_	Maximal I_Na_ conductance	√	√
G_Ks_	Maximal I_Ks_ conductance	√	√
G_K1_	Maximal I_K1_ conductance	√	√
G_CaL_	Maximal I_CaL_ conductance	√	√
G_to_	Maximal I_to_ conductance	√	√
G_Kr_	Maximal I_Kr_ conductance	√	√
G_Ncx_	Maximal I_Ncx_	√	√
G_BNa_	Maximal I_BNa_ conductance	√	√
G_BCa_	Maximal I_BCa_ conductance	√	√
G_Nak_	Maximal I_Nak_	√	√
G_PCa_	Maximal I_PCa_ conductance	√	√
G_PK_	Maximal I_PK_ conductance	√	√
G_Kur_	Maximal I_Kur_ conductance	√	√
G_rel_	Maximal I_rel_ via RyR	√	√
G_up_	Maximal I_up_ via SERCA	√	√
G_KATP_	Maximal I_KATP_ conductance	√	√
G_KAch,Ado_	Maximal I_KAch_ conductance	√	√
G_CaNa_	Maximal I_CaNa_ conductance	N/A	√
G_CaK_	Maximal I_CaK_ conductance	N/A	√
G_ClCa_	Maximal I_ClCa_ conductance	N/A	√
G_BCl_	Maximal I_BCl_ conductance	N/A	√
G_NaL_	Maximal I_NaL_ conductance	N/A	√

**Table 2 ijms-22-01265-t002:** Experimental data on AP biomarkers [58] under atrial fibrillation (AF) and sinus rhythm (SR) conditions.

AP biomarkers	SR (*n* = 238)	AF (*n* = 214)
APD_90_ (ms)	317.41 ± 43.19	217.45 ± 35.74
APD_50_ (ms)	138.09 ± 45.14	100.41 ± 6.31
RMP (mV)	−73.98 ± 0.86	−76.85 ± 0.83
dVdt_max_(V/s)	219.44 ± 14.65	231.56 ± 16.51

Note: Values are presented in mean ± standard error. AP biomarkers include APD_50_, APD_90_, RMP, and dVdt_max_. Abbreviations: dVdt_max_: Maximum upstroke velocity, RMP: Resting membrane poTable 50. and APD_90_: AP duration at 50% and 90%, respectively.

**Table 3 ijms-22-01265-t003:** Parameters associated with *Pitx2*-induced remodelling.

Experimental Observations	Parameters	Changes (%)
−60% (*SCN5A* and *SCN1B*) [22] −40% (*SCN5A* and *SCN1B*) [25] +95% *SCN5A* [17] No change [18] +70–90% (*SCN1B*) [63]	G_Na_	+10%
+500% *CACNA1D* [12] −50% *CACNA1C* [26] −30% *CACNA1C* [25] Decreased *CACNA1C* [11]	G_CaL_	+30%
+150% (*KCNQ1*) [26] Increased voltage-dependent potassium current [11] +80% (*KCNQ1*) [63] +50–70% (*KCNQ1*) [15]	G_Ks_	+150%
−20% (*KCNJ2* and *KCNJ12*) [22] +30 (*KCNJ2* and *KCNJ12*) [25] −25% *KCNK5* [18]	G_K1_	−30%
+145% *RyR2* [12] +30% *RyR2* [23] +30% *RyR2* [25] +10% *RyR2* [17]	G_rel_	+100%
+50% *ATP2A2* [12] +1000% *ATP2A2* [23] +100% *ATP2A2* [25] +12% *ATP2A2* [17] +37% *SERCA2* [24]	G_up_	+30%
−55% *GJA1* [22] −5% *GJA1* [17] +100% *GJA1* [12] −58% *GJA1* [11]	D	−50%

Note: We altered maximum conductances of ion currents according to changes in mRNA expression. Parameters associated with *Pitx2*-induced remodelling of ion channels include G_Na_, G_CaL_, G_Ks_, G_K1_, G_rel_, and G_up_. D is a scalar coefficient describing the intercellular electrical coupling via gap junctions. These parameters used in the present study are consistent with changes of Pitx2-4 in our previous study [36].

**Table 4 ijms-22-01265-t004:** Parameters associated with blocking effects of antiarrhythmic drugs.

Currents	Disopyramide (Class 1a)			Quinidine (Class 1a)			Propafenone (Class 1c)		
	IC_50_ (μM)	nH	Ref.	IC_50_ (μM)	nH	Ref.	IC_50_ (μM)	nH	Ref.
I_Na_	168.4	1.09	[64]	14.6	1.22	[64]	1.2	1.0	[65]
I_CaL_	1036.7	1.0	[64]	14.9	1.0	[66]	1.7	1.0	[67]
I_to_	20.9	1.0	[68]	21.8	1.0	[69]	4.8	1.0	[70]
I_Ks_	88.1	1.0	[71]	44.0	1.0	[72]	-	-	-
I_Kr_	14.4	0.91	[64]	0.72	1.06	[64]	2.0	1.0	[73]
I_Kur_	25.0	1.0	[74]	6.6	1.0	[69]	4.4	1.0	[75]
I_K1_	-	-	-	42.6	1.0	[69]	16.8	1.0	[76]
I_KATP_	17.8	1.0	[77]	10.0	1.0	[78]	63.1	1.0	[77]
I_KAch_	1.7	1.0	[79]	-	-	-	0.7	1.0	[79]

Note: Antiarrhythmic drugs investigated include disopyramide, quinidine, and propafenone. Ion currents associated with these drugs included I_Na_, I_CaL_, I_to_, I_Ks_, I_Kr_, I_Kur_, I_KATP_, I_KAch_, and I_K1_. Blocking effects of drugs on ion currents were modelled with the half-maximal inhibitory concentration (IC_50_) and Hill coefficient (nH) value. IC_50_ and nH were extracted from the literature.

**Table 5 ijms-22-01265-t005:** AP biomarkers obtained from the Bai et al. model and the Grandi et al. model under atrial fibrillation (AF) and sinus rhythm (SR) conditions.

Biomarkers	Bai et al. Model		Grandi et al. Model	
	SR (*n* = 745)	AF (*n* = 621)	SR (*n* = 170)	AF (*n* = 87)
APD_90_ (ms)	250.10 ± 41.59	181.93 ± 23.66	251.08 ± 61.50	241.82 ± 37.98
APD_50_ (ms)	187.84 ± 32.36	121.22 ± 24.13	66.30 ± 29.74	52.45 ± 22.18
RMP (mV)	−79.28 ± 3.29	−79.81 ± 3.22	−81.01± 0.18	−81.03 ± 0.23
dVdt_max_(V/s)	197.87 ± 83.18	219.21 ± 90.48	351.14 ± 48.90	368.66 ± 52.15

Note: Values are presented as mean ± standard error. AP biomarkers include APD_50_, APD_90_, RMP, and dVdt_max_. Abbreviations: dVdt_max_: Maximum upstroke velocity, RMP: Resting membrane potential, and APD_50_ and APD_90_: AP duration at 50% and 90%, respectively.

**Table 6 ijms-22-01265-t006:** A quantitative summary of the effects of class I antiarrhythmic drugs on human atrial electrical activity using the Bai et al. model.

Model		Cell		Tissue	
		dVdtmax(V/s)	APD90 (ms)	CV (m/s)	WL (mm)
SR		197.87 ± 83.18	250.10 ± 41.59	0.56 ±0.08	139.62 ±25.56
AF		219.21 ± 90.48	181.93 ± 23.66	0.45 ± 0.11	81.13 ± 18.91
Disopyramide	Diso_L	205.45 ± 89.15	188.29 ± 24.57	0.43 ± 0.11	80.40 ± 19.83
	Diso_M	198.77 ± 88.66	192.17 ± 26.09	0.42 ± 0.11	80.24 ± 20.71
	Diso_H	188.88 ± 87.20	199.32 ± 28.53	0.41 ± 0.11	80.63 ± 22.68
Quinidine	Quin_L	222.87 ± 91.63	206.57 ± 32.27	0.45 ± 0.11	92.56 ± 21.40
	Quin_M	218.99 ± 89.51	219.08± 36.78	0.45 ± 0.11	97.09 ± 22.51
	Quin_H	201.43 ± 80.97	235.80 ± 41.95	0.43 ± 0.10	99.70 ± 23.74
Propafenone	Prop_L	215.08 ± 91.57	196.69 ± 27.72	0.44 ± 0.11	86.22 ± 20.36
	Prop_M	215.35 ± 93.67	205.60 ± 32.13	0.44 ± 0.11	89.78 ± 21.69
	Prop_H	217.62 ± 95.29	210.37± 34.46	0.45 ± 0.11	92.24 ± 22.39

**Table 7 ijms-22-01265-t007:** A quantitative summary of the effects of class I antiarrhythmic drugs on human atrial electrical activity using the Grandi et al. model.

Model		Cell		Tissue	
		dVdtmax(V/s)	APD90 (ms)	CV (m/s)	WL (mm)
SR		351.14 ± 48.90	251.08 ± 61.50	0.77 ± 0.08	212.28 ± 51.71
AF		368.66 ± 52.15	241.82 ± 37.98	0.57 ± 0.05	136.77 ± 24.27
Disopyramide	Diso_L	367.16 ± 52.92	249.24 ± 38.67	0.56 ± 0.05	140.61 ± 24.88
	Diso_M	367.33 ± 51.76	255.12 ± 39.91	0.56 ± 0.05	143.97 ± 25.35
	Diso_H	363.94 ± 51.45	270.53 ± 42.61	0.56 ± 0.05	151.84 ± 26.99
Quinidine	Quin_L	359.86 ± 51.05	236.49 ± 44.39	0.56 ± 0.05	131.91 ± 27.76
	Quin_M	369.20 ± 51.86	241.55 ± 38.10	0.57 ± 0.05	136.75 ± 24.29
	Quin_H	310.85 ± 46.74	249.84 ± 59.37	0.51 ± 0.05	126.98 ± 31.85
Propafenone	Prop_L	367.93 ± 52.13	240.93 ± 40.27	0.57 ± 0.05	136.17 ± 25.83
	Prop_M	367.49 ± 51.74	235.57 ± 43.40	0.56 ± 0.05	133.07 ± 27.36
	Prop_H	366.40 ± 51.65	231.16 ± 45.25	0.56 ± 0.01	130.35 ± 28.21

**Table 8 ijms-22-01265-t008:** Alterations of electrophysiological properties.

Conditions	dVdtmax		APD90		CV		WL	
	Bai	Grandi	Bai	Grandi	Bai	Grandi	Bai	Grandi
SR Vs. AF	↑	↑	↓	↓	↓	↓	↓	↓
Disopyramide Vs. AF	↓	↓	↑	↑	↓	↓	↑	↑
Quinidine Vs. AF	↓	↓	↑	↑	↓	↓	↑	↓
Disopyramide Vs. AF	↓	↓	↑	↓	↓	↓	↑	↓

## Data Availability

Not applicable.

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
