# Peer review of "In Silico Assessment of Class I Antiarrhythmic Drug Effects on Pitx2-Induced Atrial Fibrillation: Insights from Populations of Electrophysiological Models of Human Atrial Cells and Tissues"

_ijms, 2021, doi:10.3390/ijms22031265_

Round 1
Reviewer 1 Report
In this study, Bai J and colleagues showed in silico assessment of class I antiarrhythmic drugs effects on pitx2-induced atrial fibrillation. They adopted 2 models: Bai et at model displaying a type-1 AP with notch-and-dome morphology and Grandi et al model displaying a type-3 AP with typical triangular shape. Simulated results showed that pitx2-induced remodeling increased dVdt max and CV, and decreased APD90 and WL, and class I AADs decreased dVdt max and CV and prolonged APD90. They also showed that quinidine and disopyramide led to WL prolongation compared to the drug-free AF conditions, but propafenone caused WL shortening; therefore, they concluded that quinidine and disopyramide might be effective against pitx2-induced AF. The manuscript is almost well-written, and may be interesting for the readers of this fields. However, I have several major concerns and questions that should be addressed by the authors. Most major concerns are the rationale for setting parameters chosen in this study, and the suitability and accuracy of the obtained results. My concerns and questions are expressed as follows.
- The authors set the ion channel parameters associated with pitx2-induced remodeling as shown in table 3. However, it is unclear how to set the parameters and whether the parameters are suitable. What is the rationale for setting each parameter? In their previous paper (PLOS comp Biol 2020), 4 different scenarios were considered for simulating pitx2-induced remodeling. The parameters in this study seem like not consistent with any scenarios.
- Gap junction and structural remodeling parameters (such as fibrosis density) were not included in this study although gap junction was included in the previous study. Because these factors have great impact on the results, especially CV and WL, I guess such parameter should be added.
- In figure 4, GK1 was positively correlated to dVdt max in Bai et al model, not consistent with the result in Grandi et al model. Moreover, GKs and Gto were not consistent between 2 models. The authors should discuss the discrepancies.
- The authors described disopyramide, quinidine, and propafenone prolonged APD90 and decreased dVdtmax and CV in a dose-dependent manner as major findings. However, disopyramide seems like not significantly different in dVdt max in Figure 5a. In addition, the authors should edit the figure 5 to 8 and table 6 so that the readers easily understand whether statistical significance is present or not.
- The authors concluded that that quinidine and disopyramide might be more effective against pitx2-induced AF by prolonging WL. However, they only showed the alterations of dVdt max, APD90, CV, and WL in response to class I AADs in pitx2-induced AF. It is better to show whether class I AAD really inhibit pitx2-induced AF using in silico model.
- This study presented simulation results using 2 models of pitx2-induced AF and the effects of class I AADs for it. Although the authors adopted 2 AP models, they didn’t fully validate the suitability and accuracy of these models. Therefore, it is difficult to judge whether this model in this study is suitable or useful for evaluating the mechanism of AF and for drug screening.
Author Response
In this study, Bai J and colleagues showed in silico assessment of class I antiarrhythmic drugs effects on pitx2-induced atrial fibrillation. They adopted 2 models: Bai et at model displaying a type-1 AP with notch-and-dome morphology and Grandi et al model displaying a type-3 AP with typical triangular shape. Simulated results showed that pitx2-induced remodeling increased dVdt max and CV, and decreased APD90 and WL, and class I AADs decreased dVdt max and CV and prolonged APD90. They also showed that quinidine and disopyramide led to WL prolongation compared to the drug-free AF conditions, but propafenone caused WL shortening; therefore, they concluded that quinidine and disopyramide might be effective against pitx2- induced AF. The manuscript is almost well-written, and may be interesting for the readers of this fields. However, I have several major concerns and questions that should be addressed by the authors. Most major concerns are the rationale for setting parameters chosen in this study, and the suitability and accuracy of the obtained results. My concerns and questions are expressed as follows.
- The authors set the ion channel parameters associated with pitx2-induced remodeling as shown in table 3. However, it is unclear how to set the parameters and whether the parameters are suitable. What is the rationale for setting each parameter? In their previous paper (PLOS comp Biol 2020), 4 different scenarios were considered for simulating pitx2-induced remodeling. The parameters in this study seem like not consistent with any
Reply 1: We would like to thank this reviewer for raising this issue and apologize for the confusion of parameters used in the present study. Firstly, Pitx2-induced electrical remodelling is characterized by changes in the mRNA levels corresponding to ion channels in biological experiments, therefore we assumed that these changes in mRNA expression are quantitatively reflected at the final functional level of ion channels and incorporated alterations in the maximal conductances of ionic currents into the cellular models. This (i.e., mRNA alterations often do not match electrophysiological alterations) is one of the main limitations and had been discussed in the limitation section. Secondly, it was found that identified ionic targets and their remodelled degree are varied in different experiment studies. Thus, populations of sampled models of human atrial electrophysiology were generated to capture intersubject variability. These parameters were independently varied following a log-normal distribution and sigma was set to be 0.2 to cover a range of variability similar to that seen in experiments based on previous studies. Finally, in our previous study (PLOS comp Bio 2020), four different scenarios (Pitx2-1, Pitx2-2, Pitx2-3 and Pitx2-4) were considered to capture variations in identified ionic targets. Pitx2-1, Pitx2-2 and Pitx2-3 are respectively, corresponding to experimental data of three research groups, whereas Pitx2- 4 took into account all identified regulators and best represented Pitx2-induced electrical remodeling. Therefore, the parameters used in the present study are consistent with changes of Pitx2-4 in our previous paper (PLOS comp Bio 2020).
Change in Text:
Page 4, lines 115-116: These parameters used in the present study are consistent with changes of Pitx2-4 in our previous study [36].
Page 17, lines 579-583: Secondly, we assumed that these changes in mRNA expression are quantitatively reflected at the final functional level of ion channels to obtain human AF
myocytes models that reproduced the experimentally observed changes in the mRNA levels corresponding to key proteins under Pitx2-induced electrical remodelling conditions. Of note, mRNA alterations often do not match electrophysiological alterations. And this is one of the main limitations.
- Gap junction and structural remodeling parameters (such as fibrosis density) were not included in this study although gap junction was included in the previous study. Because these factors have great impact on the results, especially CV and WL, I guess such parameter should be
Reply 2: We would like to thank this reviewer for his or her detailed suggestions. In the revised MS, structural remodeling parameters included in our previous study were considered in the present study. Results associated with CV and WL have been updated. Change in Text:
Tables 3, 6 and 7 and Figures 5 and 6 have been updated.
- In figure 4, GK1 was positively correlated to dVdt max in Bai et al model, not consistent with the result in Grandi et al model. Moreover, GKs and Gto were not consistent between 2 models. The authors should discuss the
Reply 3: We would like to thank this reviewer for raising this issue. In the revised MS, discrepancies between Bai et al and Grandi et al models have been discussed.
Change in Text:
Page 14, lines 408-429: The Bai et al. model was developed by taking into account ionic differences between atrial and ventricular cells based on the previous human ventricular model (TP model) developed by ten Tusscher and Panfilov [91]. TP model included a subspace calcium variable that controls the dynamics of the ICaL and the ryanodine receptor current. The phenomenological description of ICaL-induced calcium release was used with a reduced version of the ryanodine receptor Markov model developed by Stern et al.[92]and Shannon et al.[93]. The AP profile of the Bai et al. model is a spike-and- dome-type action potential and is comparable to the Courtemanche et al. model[94]. The Grandi model was developed largely based on their previous human ventricular model, and therefore the formulation of transmembrane currents differs significantly from the Bai et al. and Courtemanche et al. models. In addition to the main transmembrane currents, the Grandi atrial model also includes the formulation of two chloride and a potassium plateau current. The calcium subsystem model is based on the one in the rabbit ventricular model by Shannon et al.[93]. Therefore, the calcium transient of the Grandi et al. model is comparable to the Bai et al. model, but the Grandi et al. model with a triangular AP shape is different from the Bai et al. model. This difference can be reflected by the effects of ion currents on depolarization and repolarization. Consistent with this study [95], the excitability of the Bai et al. model is modulated by both INa and IK1, whereas that of the Grandi model is regulated only by INa. During the repolarization duration, Ito and IKur regulate the AP shape (the notch) at phase 1 and IKs and IKr are the important potassium currents that regulate APD90. These characteristics are derived from those of the base model (TP model). However, there is no notch shape in the AP of the Grandi
model and thereby all potassium currents (including Ito and IKur) are modulated APD90. Among these potassium currents, IKs, IKach and IKATP have small effects on APD90.
- The authors described disopyramide, quinidine, and propafenone prolonged APD90 and decreased dVdtmax and CV in a dose-dependent manner as major findings. However, disopyramide seems like not significantly different in dVdt max in Figure 5a. In addition, the authors should edit the figure 5 to 8 and table 6 so that the readers easily understand whether statistical significance is present or
Reply 4: We would like to thank this reviewer for important suggestions. We have revised figures to show statistical significance of dVdtmax, APD90 (revised Figure 5), CV and WL (revised Figure 6) according to the reviewer’s suggestions. In addition, values of these biomarkers are listed in Tables 6 and 7.
Change in Text:
Tables 6 and 7 and Figures 5 and 6 have been updated.
- The authors concluded that that quinidine and disopyramide might be more effective against pitx2-induced AF by prolonging However, they only showed the alterations of dVdt max, APD90, CV, and WL in response to class I AADs in pitx2-induced AF. It is better to show whether class I AAD really inhibit pitx2-induced AF using in silico model.
Reply 5: We would like to thank this reviewer for his/her comments. Whole heart simulations with complex models and large parameter dimensions are computationally expensive and generally require high performance computing resources; running millions (or more) of simulations to obtain estimates of model outputs is likely to be prohibitively expensive. Therefore, we didn’t investigate whether class I AAD really inhibit pitx2-induced AF using populations of 3D atrial models. We agree that further studies using populations of 3D atrial models should be conducted when conditions permit. In the revised MS, this limitation has been included in the discussion section.
Change in Text:
Page 18, lines 625-630: Finally, further studies using populations of 3D atrial models should be conducted to show whether class I AAD really inhibit pitx2-induced AF. However, whole heart simulations with complex models and large parameter dimensions are computationally expensive and generally require high performance computing resources; running millions (or more) of simulations to obtain estimates of model outputs is likely to be prohibitively expensive. Further studies will be performed if conditions permit.
- This study presented simulation results using 2 models of pitx2-induced AF and the effects of class I AADs for it. Although the authors adopted 2 AP models, they didn’t fully validate the suitability and accuracy of these models. Therefore, it is difficult to judge whether this model in this study is suitable or useful for evaluating the mechanism of AF and for drug screening.
Reply 6: We would like to thank this reviewer for raising this important question. Firstly, due to the lack of experimental data on humans, the electrophysiological representation of
AF-induced remodelling in the human atrial AP model is based on data from previous mice models of Pitx2-induced AF. Secondly, although shortened atrial repolarization, a more depolarized resting membrane potential and abnormalities in calcium cycling were observed in atrial cardiomyocytes of mice with reduced Pitx2 mRNA and these models (including Bai et al. and Grandi et al. models) can reproduce these phenomena in our previous study, Pitx2-induced AP data of human atrial cardiomyocytes is not available to validate the suitability and accuracy of our models so far. In fact, these models were used to predict Pitx2-induced APs and effects of class I AADs for it. In the revised MS, we have addressed this limitation and have revised inappropriate statements in the full text.
Change in Text:
Pages 17-18, lines 587-595: Firstly, the electrophysiological representation of AF- induced remodelling in the human atrial AP model is based on data from previous mice models of Pitx2-induced AF, however, because of the lack of experimental data on humans. Although shortened atrial repolarization, a more depolarized resting membrane potential and abnormalities in calcium cycling were observed in atrial cardiomyocytes of mice with reduced Pitx2 mRNA and these models (including Bai et al. and Grandi et al. models) can reproduce these phenomena in our previous study[27], Pitx2-induced AP data of human atrial cardiomyocytes is not available to further validate the suitability and accuracy of our models so far. Special attention must be paid to the differences between mice and human atrial cells [98].

Reviewer 2 Report
The paper is well written and deals with an interesting topic. Therefore, the authors´ results seems to be interesting for the scientific community. Nevertheless, I would have some questions/suggestions regarding the authors´ work:
- In the absence of contraindication, class I antiarrhythmics are established and commonly applied drugs in AF patients. However, since class I antiarrhythmics seem to be specifically efficient in the Pitx2-induced AF, I would be interested in the potential differences in electrophysiological remodeling when compared to other forms of AF. Of note, the authors only evaluated the difference between Pitx2-induced AF and SR. Is there any possibility to investigate this issue?
- With regard to the results obtained for quinidine: It seems to have the highest impact on APD prolongation which is known to affect QT interval with increased risk for reentry mechanisms, if the same effect would appear in the ventricle myocardium. I would recommend discussing this issue.
- As already mentioned by the authors as a limitation, the human atrial AP model is based on data from previous mice models of Pitx2-induced AF. Could the authors specify the differences and potential clinical implications with regards to the model application. Of note, in contrast to human AP, which is mostly based on Ica, in rodents AP is mostly driven by the Ito.
- The authors´ calculations were driven by results obtained from mRNA levels of relevant ionic currents. Of note, mRNA alterations often do not match electrophysiological alterations. This should be specified as one of the main limitations.
- I would recommend drawing a table which represents specific alterations of electrophysiological properties win AF and with call I antiarrhythmics application in comparison to SR (APD prolonged, wavelenght increased etc.)
Author Response
The paper is well written and deals with an interesting topic. Therefore, the authors´ results seems to be interesting for the scientific community. Nevertheless, I would have some questions/suggestions regarding the authors´ work:
- In the absence of contraindication, class I antiarrhythmics are established and commonly applied drugs in AF patients. However, since class I antiarrhythmics seem to be specifically efficient in the Pitx2-induced AF, I would be interested in the potential differences in electrophysiological remodeling when compared to other forms of AF. Of note, the authors only evaluated the difference between Pitx2-induced AF and SR. Is there any possibility to investigate this issue?
Reply 1: We would like to thank this reviewer for his/her comments. We agree that electrophysiological remodelling is different for different forms of AF and effects of class I AADs on different forms of AF should be compared. In the revised MS, we compared electrical remodelling of Pitx2-induced AF with those of other forms of AF. We also discussed studies on effects of class I AADs on different forms of AF.
Changes in Text:
Pages 13-14, lines 340-414: AF is the most common cardiac arrhythmia with well- established clinical and genetic risk components. AF can be broadly divided into two types, genetic and acquired types according to the factors that cause AF. Acquired AF usually begins in a self-terminating paroxysmal form (pAF) and this pattern often evolves to become chronic form (cAF) [1]. In addition to acquired AF, genetic factors are presumed as key in the development of AF, especially in familial cases (fAF) without cardiac pathology. The total genetic contribution to fAF risk can be broadly divided into three components: rare coding variation, common variation and undiscovered variation [91].
For common variation, the genetic loci associated with fAF were first identified and are located on chromosome 4q25, upstream of the transcription factor gene Pitx2 [10]. Pitx2 deficiency resulted in electrical and structural remodelling, and impaired repair of the heart in murine models [18]. Then, meta-analysis of AF cases identified a novel locus for fAF (ZFHX3, rs2106261)[3]. Furthermore, a meta-analysis of multiple well-phenotyped GWAS identified six new susceptibility loci for fAF, including 1q24 (PRRX1), 7q3 (CAV1), 14q23 (SYNE2), 9q22 (FBP1 and FBP2), 15q24 (HCN4), and 10q22 (upstream of SYNPO2L and MYOZ1)[5]. By meta-analyses of SNP-associations with AF, researchers discovered five novel loci near the genes NEURL (rs12415501 and rs6584555), GJA1 (rs13216675), TBX5 (rs10507248), CAND2 (rs4642101), and CUX2 (rs6490029)[9]. Five GWAS were conducted in 2017, with a total of no more than 30 novel loci identified. Roselli et al. conducted the largest GWAS-meta-analysis, and found that there were 97 loci significantly associated with AF [8]. Nielsen et al. found 142 independent risk variants at 111 loci and prioritized 151 functional candidate genes likely to be involved in fAF. Of these, 80 loci have not been previously reported [92]. Among these common genes, the links between fAF and Pitx2 or TBX5 have been deeply investigated. Experimental studies demonstrated that TBX5 directly activated Pitx2, and TBX5 and Pitx2 antagonistically regulated membrane effector genes SCN5A, GJA1, RYR2, DSP, and ATP2A2[17].
For rare coding variation, S140G in KCNQ1 gene was firstly identified [93]. The potassium voltage-gated channel subfamily E genes encode the regulatory β-subunits of the channels producing the delayed-rectifier potassium current. Gain-of-function mutations in KCNE1, KCNE2, KCNE3, and KCNE5 have been associated with fAF. Furthermore, the genes (i.e., KCNA5, KCND3, and KCNH2) coding α-subunit of voltage-gated potassium channels Kv1.5, Kv4.3, and Kv11.1, and α-subunit of inwardly rectifying potassium channels Kir2.1, Kir3.4, and Kir6.1 (i.e., KCNJ2, KCNJ5, and KCNJ8) also showed significant associations with fAF risk. fAF-associated potassium channel variants have a gain of channel function, with an expected shortening of the atrial action potential duration and atrial refraction period. The fAF-associated sodium channel genes included SCN5A, SCN10A, and genes coding sodium voltage-gated channel β subunit 1–4 (SCN1B, SCN2B, SCN3B, and SCN4B). The mutations in SCN5A exhibited compromised peak sodium current and an increased sustained sodium current. The related variations in SCN10A were implicated in the modulation of late sodium current, while mutations in SCN1B-4B were engaged with modulation of inward sodium current. Several non-ion channel genes did not directly alter the action potential, but instead would be expected to instigate the onset of fAF through alternative mechanisms. These included NPPA, NUP155, LMNA, GJA5, AGT, and ACE [94].
For pAF, APD90, ICaL and INcx were unaltered, indicating the absence of AF-induced electrical remodeling [95]. In contrast, there were increases in SR Ca2+ leak and incidence of delayed after-depolarizations in pAF. Ca2+ -transient amplitude and sarcoplasmic reticulum Ca2+ load were larger in pAF. Ca2+ -transient decay was faster in pAF, but the decay of caffeine- induced Ca2+ transients was unaltered, suggesting increased SERCA2a function. In agreement, phosphorylation (inactivation) of the SERCA2a-inhibitor protein phospholamban was increased in pAF. Ryanodine receptor fractional phosphorylation was unaltered in pAF, whereas ryanodine receptor expression and single-channel open probability were increased. Increased diastolic sarcoplasmic reticulum Ca2+ leak and related delayed after-depolarizations promote cellular arrhythmogenesis in pAF patients. Biochemical, functional, and modeling studies point to a combination of increased sarcoplasmic reticulum Ca2+ load related to phospholamban hyperphosphorylation and ryanodine receptor dysregulation as underlying mechanisms.
In addition to Ca2+-handling, cAF involves electrophysiological and structural remodeling.
Differences in cardiomyocyte Ca2+-handling properties between patients with pAF and cAF point towards progressive changes in atrial Ca2+-handling due to AF itself[96]. Ca2+- transient amplitude is increased in pAF, but decreased in cAF[97]. Consistent with its role as a frequency sensor, CaMKII autophosphorylation and CaMKII-dependent RyR2 phosphorylation are elevated in cAF but not pAF. Rate induced Ca2+-loading causes enhanced binding of Ca2+ to calmodulin, which activates the phosphatase calcineurin. Calcineurin then dephosphorylates the nuclear factor of activated T-cells, which translocates into the nucleus and inhibits production of CACNA1C mRNA, decreasing the message for the ICaL α-subunit, thereby reducing its protein expression and ion-transport function[98]. In parallel, nuclear factor of activated T-cells suppresses the production of microRNA (miR)-26 by binding to and negatively regulating sites upstream to the transcriptional start site in human and mouse atrium[99]. One of the binding targets of the miR-26 seed site is KCNJ2, the gene encoding the IK1 channel. Reduced miR-26 expression caused by AF removes miR-26-induced destabilization of the KCNJ2 message and inhibition of its translation. Inward-rectifier current functional expression is enhanced by this mechanism, as well as by increased IKACh in human and canine models[100]. Recent work also shows that NLRP3 activation increases the gene expression of the channels subunits underlying the atrial-selective currents IKur and IKACh [101]. In addition to these pathways, protein-kinase isoform switches upregulate the constitutive activity of IKACh, and upregulation of 2-pore and Ca2+-dependent K+-channels[102]. In addition to AF-induced electrical remodelling, rapid cardiomyocyte firing leads to fibroblast activation via a diffusible substance in HL-1 atrial-derived cardiomyocytes, which seems to be ROS- derived from cardiomyocyte nicotinamide adenine dinucleotide phosphate oxidase stimulated by Ca2+-loading[103]. ROS are known to activate NLRP3-inflammasomes in other systems and NLRP3-inflammasome activation is known to cause atrial fibrosis[104].
- With regard to the results obtained for quinidine: It seems to have the highest impact on APD prolongation which is known to affect QT interval with increased risk for reentry mechanisms, if the same effect would appear in the ventricle myocardium. I would recommend discussing this
Reply 2: We would like to thank this reviewer for important suggestions. In the revised MS, we have discussed this issue.
Changes in Text:
Pages 15-16, lines 481-488: However, excessive prolongation of the electrocardiographic QT interval by drugs is associated with the occurrence of a potentially lethal form of polymorphic ventricular tachycardia termed torsades de pointes. A previous study showed that quinidine causes greater QT prolongation in women than in men at equivalent serum concentrations. This difference may contribute to the greater incidence of drug‐induced torsades de pointes observed in women taking quinidine and has implications for other cardiac and noncardiac drugs that prolong the QTc interval. Adjustment of dosages based on body size alone are unlikely to substantially reduce the increased risk of torsades de pointes in women[105].
- As already mentioned by the authors as a limitation, the human atrial AP model is based on data from previous mice models of Pitx2-induced AF. Could the authors specify the differences and potential clinical implications with regards to the model application. Of note, in contrast to human AP, which is mostly based on Ica, in rodents AP is mostly driven by the
Reply 3: We would like to thank this reviewer for important suggestions. In the revised MS, the differences between human and mice AP and potential clinical implications of this human AP models developed with these data of mice models were specified in the limitation section.
Page 17, lines 553-567: Special attention must be paid to the differences between mice and human atrial cells [113]. In general, APD is approximately 50 ms in mouse, compared to 250 ms in humans. The AP morphology reflects the contribution of numerous depolarizing and repolarizing currents. Even when the same type of ion channel is expressed in human and mice, its contribution to the AP morphology may differ substantially, given the large difference in APD. In the mouse heart, ICaL contributes less to the AP than in humans and therefore the murine AP shows a gradual repolarization rather than a distinct plateau phase. The much faster repolarization in mice is mediated by transient outward K+ currents with a fast and slow recovery from inactivation, a slowly inactivating K+ current and a non-inactivating, steady state current. In humans, the transient outward K+ current with a fast recovery is mainly involved in phase 1 repolarization, with more prominent expression in the epicardium. In addition, the rapid and slow delayed outward rectifier K+ currents are predominantly responsible for phase 3 repolarization. However, studies in mice did observe the rapid and slow delayed rectifier K+ currents, but their contribution to repolarization under physiological conditions is probably negligible or minor[114].
- The authors´ calculations were driven by results obtained from mRNA levels of relevant ionic currents. Of note, mRNA alterations often do not match electrophysiological alterations. This should be specified as one of the main
Reply 4: Thanks for comments of this reviewer. According to this suggestion, we have added this limitation in the discission section.
Change in text:
Page 17, lines 572-576: Secondly, we assumed that these changes in mRNA expression are quantitatively reflected at the final functional level of ion channels to obtain human AF myocytes models that reproduced the experimentally observed changes in the mRNA levels corresponding to key proteins under Pitx2-induced electrical remodelling conditions. Of note, mRNA alterations often do not match electrophysiological alterations. Andy this is one of the main limitations.
- I would recommend drawing a table which represents specific alterations of electrophysiological properties win AF and with call I antiarrhythmics application in comparison to SR (APD prolonged, wavelenght increased )
Reply 5: We would like to thank this reviewer for this suggestion. In the revised MS, we added a new table for summering specific alteration of AF and with drug application in comparison to SR.
Change in text: A new table (Table 8) has been added.

Round 2
Reviewer 2 Report
The authors´ revision is sufficient and addressed my major concerns.
Author Response
The authors´ revision is sufficient and addressed my major concerns.
Reply:We thank this Reviewer for his/her overall positive evaluation of our study.
